# Majority of the Bests: Improving Best-of-N via Bootstrapping

**Amin Rakhsha**[1,2,3*]      **Kanika Madan**[3]      **Tianyu Zhang**[3]

**Amir-massoud Farahmand**[4,5,1]      **Amir Khasahmadi**[3]

[1]University of Toronto    [2]Vector Institute    [3]Autodesk

[4]Polytechnique Montréal    [5]Mila - Quebec AI Institute

## Abstract

Sampling multiple outputs from a Large Language Model (LLM) and selecting the most frequent (Self-consistency) or highest-scoring (Best-of-N) candidate is a popular approach to achieve higher accuracy in tasks with discrete final answers. Best-of-N (BoN) selects the output with the highest reward, and with perfect rewards, it often achieves near-perfect accuracy. With imperfect rewards from reward models, however, BoN fails to reliably find the correct answer and its performance degrades drastically. We consider the distribution of BoN's outputs and highlight that, although the correct answer does not usually have a probability close to one under imperfect rewards, it is often the most likely outcome. This suggests that the mode of this distribution can be more reliably correct than a sample from it. Based on this idea, we propose Majority-of-the-Bests (MoB), a novel selection mechanism that estimates the output distribution of BoN via bootstrapping and selects its mode. Experimental results across five benchmarks, three different base LLMs, and two reward models demonstrate consistent improvements over BoN in 25 out of 30 setups. We also provide theoretical results for the consistency of the bootstrapping. MoB serves as a simple, yet strong alternative to BoN and self-consistency, and more broadly, motivates further research in more nuanced selection mechanisms.[2]

## 1 Introduction

Scaling the inference-time computation of language models has led to a significant improvement of their performance on a variety of tasks (Brown et al., 2024; Snell et al., 2025; Wu et al., 2024; OpenAI, 2024; DeepSeek-AI, 2025). A growing number of methods have been introduced in this paradigm, such as generating long chains-of-thought (Wei et al., 2022; Muennighoff et al., 2025), asking the model to evaluate and improve its own outputs (Madaan et al., 2023), and tree search (Yao et al., 2023; Hao et al., 2023; Zhang et al., 2024). Another family of such algorithms, termed *sample-and-marginalize* by Wang et al. (2022), generate multiple outputs from the model and then aggregate them into a final answer. Examples include Self-consistency (Wang et al., 2022), Best-of-N (Lightman et al., 2023), and Weighted Best-of-N (Li et al., 2022). These methods have gained popularity due to their simplicity and scalability.

---

*Work done during internship at Autodesk. Correspondence to: `aminr@cs.toronto.edu`.

[2]Code and data available at https://github.com/arakhsha/mob

39th Conference on Neural Information Processing Systems (NeurIPS 2025).

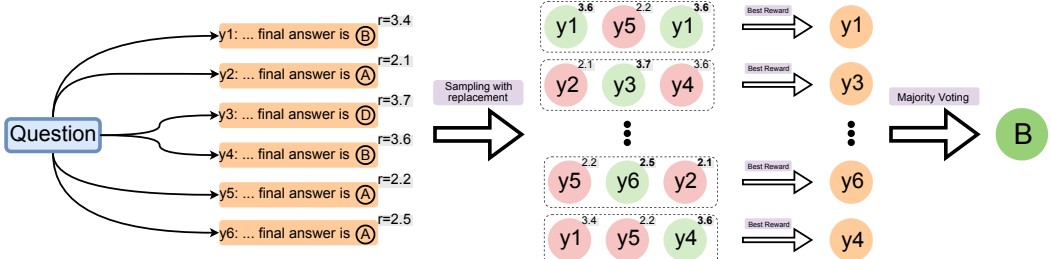

Figure 1: Majority-of-the-Bests: first, $N$ outputs are generated for the given question. Then, we create a large number of subsets of size $m < N$ by sampling with replacement from the generated outputs. From each subset, we choose the output with the highest reward. The most frequent answer among these chosen outputs is reported as the final answer.

Self-consistency (SC) (Wang et al., 2022), also referred to as "majority voting", is a widely used algorithm in this paradigm. It samples multiple outputs from the model and selects the final answer that appears most frequently among them. SC improves the performance by leveraging a key property of the model's output distribution: on difficult problems, the probability of generating the correct answer is often far from 1, making single-sample predictions unreliable. SC capitalizes on the fact that, even if the model's output distribution is imperfect, it may still favor the correct answer and generate it more frequently than incorrect ones.

Best-of-N (BoN) (Lightman et al., 2023) uses a reward model to evaluate the generated outputs and chooses the final answer in the highest-scoring output. With an ideal reward model, BoN succeeds as long as one of the generated outputs is correct. In this paper, we highlight that in the realistic setting of an imperfect reward model, the success of BoN is no longer (nearly) guaranteed. In such cases, BoN exhibits stochastic behavior akin to the underlying generative model. While the reward model improves the likelihood of selecting the correct answer, it often falls short of ensuring certainty. This is the same property that underlies the effectiveness of SC. Motivated by this observation, we show that applying a similar principle—aggregating multiple samples to identify the most probable answer—leads to a better performance over BoN.

We introduce Majority-of-the-Bests (MoB), a method that leverages bootstrapping to improve upon BoN *by approximating the most probable output of BoN*. As illustrated in Figure 1, after obtaining multiple (parallel) solution samples for a given question and computing their rewards, we apply bootstrapping: we create subsets of size $m$ by sampling with replacement from the generated outputs. For each subset, we select the sample with the highest reward. This results in a new set of high-reward samples, over which we perform majority voting to determine the final answer. Just like BoN and SC, MoB can be applied independent of the output generation procedure. It only modifies the selection of the final answer with marginally extra computation on the CPU. We provide a procedure to adaptively select $m$, eliminating any critical hyperparameters from the algorithm. We show the consistency of the algorithm theoretically, and empirically show significant improvements over BoN on 25 out of 30 tested setups.

## 2 Background

In this section, we formulate BoN and bootstrapping and provide some background for the algorithm and its theoretical grounds. Given a prompt $x$, in the standard procedure with LLMs, we sample an output $Y \sim p_{\text{ref}}$ from a base model $p_{\text{ref}}$. This output yields a corresponding final answer $Z = f(Y)$ after applying a post-processing or evaluation function $f$. For example, for a multiple choice problem, $Z$ is the chosen option and $Y$ is the whole output containing both $Z$ and its justification. We denote the distribution of the final answer in this procedure as $\pi_{\text{ref}}$, that is, $Z \sim \pi_{\text{ref}}$. The goal is to find the correct final answer $z^*$. We define the success probability for this given problem as the probability of selecting the correct final answer. If the algorithm's final answer is $Z$, the success probability is defined as $\mathbb{P}(Z = z^*)$. Given a dataset of questions, the average of the success probabilities over all questions is referred to as the accuracy. For the standard procedure, the success probability is equal to $\pi_{\text{ref}}(z^*)$ and the corresponding accuracy is called the *pass@1* accuracy. We assume access to a

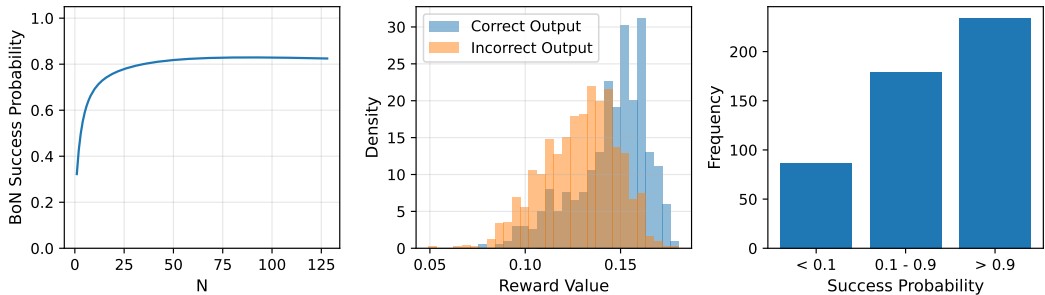

Figure 2: *(Left)* BoN's success probability as a function of $N$ for question 647 from MMLU-Pro-Math. The success probability remains below 80%. *(Middle)* Distribution of the reward for correct and incorrect outputs for the same question. A separation between the two distributions is ideal. *(Right)* Histogram of Best-of-64 success probabilities over 500 questions.

reward model $r$ that assigns a reward $R = r(Y)$ to the output $Y$, reflecting its accuracy, coherence, or alignment with human preferences (Uesato et al., 2022; Lightman et al., 2023). For a given budget $N$, sample-and-marginalize algorithms generate $N$ independent outputs $Y_1, \ldots, Y_N \sim p_{\text{ref}}$ and select the final answer reached by one of these outputs. BoN selects the final answer from the output with the highest reward, that is,

$$Z_N^{\text{Best}} = f\big(\operatorname*{argmax}_{y \in \{Y_1, \ldots, Y_N\}} r(y)\big).$$

Regularized versions of BoN have also been introduced to address its reward hacking issues in the presence of inaccurate rewards (Jinnai et al., 2024; Ichihara et al., 2025). Alternatively, self-consistency or majority voting selects the final answer that occurs most frequently among $Z_1, \ldots, Z_N$, where $Z_i = f(Y_i)$ is the final answer for output $Y_i$. If $N$ is large enough, this most frequent answer will be the mode of the final answer distribution $\pi_{\text{ref}}$. Li et al. (2022) suggested the Weighted Best-of-N (WBoN) selection method. For each final answer, WBoN sums the rewards of all outputs that lead to it. Then, it selects the final answer with the highest total reward.

**Bootstrapping.** Bootstrapping is a powerful and widely used non-parametric resampling technique for estimating the distribution of a statistic by repeatedly drawing samples with replacement from the original dataset (Efron, 1992; Efron and Tibshirani, 1994). The core idea is to generate multiple "bootstrap samples", by sampling observations uniformly and with replacement. For each bootstrap sample, the statistic of interest is computed. The collection of these computed statistics from the many bootstrap samples forms an empirical approximation of the statistic's true distribution. We use this technique to approximate the distribution of BoN's output.

## 3 Motivation: Output Distribution of Best-of-N

To motivate our algorithm, we highlight the behavior of BoN's final answer distribution. We denote this distribution by $\pi_N$. It means,

$$Z_N^{\text{Best}} \sim \pi_N.$$

Assume among the $N$ sampled outputs, $N_c$ outputs $\{Y_1^c, \ldots, Y_{N_c}^c\} \subseteq \{Y_i\}_{i=1}^N$ yield the correct final answer: $f(Y_i^c) = z^*$. Conversely, $N_w = N - N_c$ outputs $\{Y_1^w, \ldots, Y_{N_w}^w\} \subseteq \{Y_i\}_{i=1}^N$ lead to an incorrect solution. Then, BoN's output is correct if the highest reward among the correct outputs is larger than the highest reward among the incorrect ones. Formally, we can express this condition as:

$$\max\big(r(Y_1^c), \ldots, r(Y_{N_c}^c)\big) > \max\big(r(Y_1^w), \ldots, r(Y_{N_w}^w)\big). \tag{1}$$

There are two factors that influence the probability of this event. First, note that each side of (1) is the maximum of some random variables. As the number of random variables increases, the probability distribution of their maximum shifts towards higher values. Therefore, larger values of $N_c$ and smaller values of $N_w$, make condition (1) more likely. The values of $N_c$ and $N_w$ depend on $\pi_{\text{ref}}(z^*)$,

the probability of the correct answer $z^*$ in the base model's final answer distribution $\pi_{\text{ref}}$. For large enough $n$, we will have

$$N_c \approx N \cdot \pi_{\text{ref}}(z^*) \quad , \quad N_w \approx N \cdot (1 - \pi_{\text{ref}}(z^*)).$$

It means that if the base model has a higher chance of solving the problem, BoN is also more likely to select the correct answer.

The second factor is the distribution of $r(Y_i^c)$ and $r(Y_i^w)$ on each side of (1). The reward of a correct output follows the conditional distribution $\mathcal{P}_c \triangleq \mathbb{P}(r(Y)|f(Y) = z^*)$ while the reward of an incorrect output follows the conditional distribution $\mathcal{P}_w \triangleq \mathbb{P}(r(Y)|f(Y) \neq z^*)$. We hope that the reward model assigns higher rewards to correct outputs, and $r(Y_i^c) \sim \mathcal{P}_c$ on the left side of (1) generally be larger than $r(Y_i^w) \sim \mathcal{P}_w$ on the right side.

Therefore, the success probability of BoN heavily depends on the separation between $\mathcal{P}_c$ and $\mathcal{P}_w$. A perfect reward model would always assign a higher value to a correct output than to an incorrect one. In that case, as long as at least one correct output is generated (which is highly likely for large enough $N$), condition (1) is satisfied. The resulting success probability is close to 1, indicating a nearly deterministic final answer. On the other hand, consider the case where $\mathcal{P}_c$ and $\mathcal{P}_w$ are identical. In this case, the reward of an output becomes independent of its correctness, and choosing according to the reward model will be no better than a random choice. Consequently, the success probability of BoN will be the same as the base model, i.e. $\pi_N(z^*) = \pi_{\text{ref}}(z^*)$. In practice, our reward models exhibit a middle ground between these two extremes. They might not be perfect for BoN to succeed with a single correct output, but they can still be somewhat informative to increase the success probability of BoN compared to the base model.

In Figure 2, we show an example of these dynamics for Question 647 of the MMLU-Pro-Math benchmark (Wang et al., 2024b) with base model Qwen2.5-3B (Qwen Team, 2024) and reward model ArmoRM (Wang et al., 2024a). We approximate the output distribution $p_{\text{ref}}$ with a large pool of 1400 samples. In Question 647 (Figure 2), the two distributions $\mathcal{P}_c$ and $\mathcal{P}_w$ are overlapping, and even with large values of $N$, the success probability remains below $80\%$. Nonetheless, BoN still outperforms the base model, which is equivalent to Best-of-1 and has a success probability of $30\%$ in this case.

We expect the stochasticity of BoN's output to depend on the difficulty of the question relative to the base and reward models' capabilities. For more difficult questions, the base model generates fewer correct outputs, and the reward model is less likely to distinguish the correct outputs from the incorrect ones. Through the two factors discussed above, BoN is not able to pick the correct answer with high certainty. The right plot in Figure 2 shows the histogram of the success probability of Best-of-64 among 500 randomly selected MMLU-Pro-Math problems. We see that for approximately 175 problems, BoN has a success probability between 0.1 and 0.9. That means, BoN has a significant chance of returning the correct answer but fails to do so reliably. *The idea behind our introduced method, MoB, is that if we can find the most probable output of the BoN distribution, we may reliably pick the correct answer even if its probability is well below* 1.

# 4 Majority-of-the-Bests

In Section 3, we showed that BoN's final answer is stochastic, and this stochasticity might remain true even with a very large budget $N$. In this section, we introduce Majority-of-the-Bests (MoB). MoB can select the correct answer with high probability as long as the correct answer is the most probable output of BoN, even if its probability is well below 1. We first showcase this idea in the hypothetical case where BoN's output distribution $\pi_N$ is given by an oracle. Later, we show how to estimate this distribution using bootstrapping.

## 4.1 MoB with Oracle Access to BoN's Output Distribution

Suppose the distribution of BoN's final answer $\pi_N$ is known through an oracle. Instead of sampling from this distribution, which is equivalent to BoN and is a noisy decision, we propose selecting the mode of this distribution. That is

$$z_N^{\text{OracleMoB}} = \underset{z}{\arg\max} \, \pi_N(z). \tag{1}$$

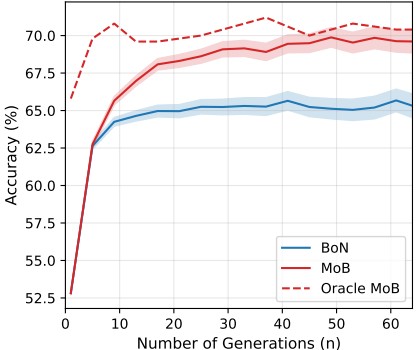 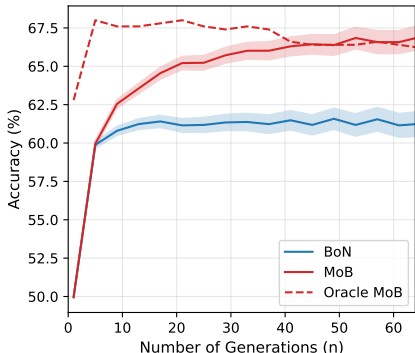

Figure 3: Final answer accuracy comparison of BoN, MoB, and Oracle MoB on MMLU-Pro-Math using Qwen2.5-3B *(Left)* and Llama3.1-8B *(Right)* as the base model, and ArmoRM as the reward model. Results are averaged across all problems and multiple runs. Shaded area indicates the standard error.

We refer to this algorithm as *Oracle MoB* as it relies on an oracle. By selecting the mode, if the correct answer has a higher probability than any of the other answers, it will be selected without any randomness that would reduce the success probability. Since $\pi_{\text{ref}} = \pi_1$, we can say SC for a large $N$ is equivalent to Oracle MoB with $N = 1$. It has been extensively shown that SC improves the LLM's original accuracy. As we will also empirically show, MoB similarly increases the accuracy of BoN by selecting the mode of its output distribution.

In Figure 3, we compare the accuracy of Oracle MoB with BoN on MATH500 (Lightman et al., 2023; Hendrycks et al., 2021) and math problems of MMLU-Pro (Wang et al., 2024b). We use the same output pool, base model, and reward model as Figure 2. We can see that depending on the value of $N$, Oracle MoB provides 5 to 10 percentage points improvement in accuracy. Oracle MoB unrealistically requires an oracle access to $\pi_N$. Next, we will show how $\pi_N$ can be estimated via bootstrapping and remove the oracle dependence.

## 4.2 MoB with Estimated BoN's Output Distribution

We now discuss how, without the oracle access to the BoN's output distribution $\pi_N$, one can approximately find its most probable output. The most obvious approach is to follow the same procedure as SC. For some $k \geq 1$, we can run $k$ independent BoN procedures, each with $m$ outputs. Then, out of the $k$ resulting answers, we select the final answer that appears the most number of times. The answer of the BoN procedures let us approximate $\pi_m$, and selecting the most frequent answer among them will approximate Oracle MoB (1) with budget $m$. We refer to this algorithm as "BoN+SC" due to its simple combination of BoN and SC. To keep the generation budget fixed at $N$, we are forced to use a smaller budget $m$ for each of the BoN runs. For now, we treat the choice of $m$ as a hyperparameter, but will return to this choice later. Assume $m < N$ and $k = \lfloor N/m \rfloor$. Formally,

$$Z_m^{\text{Best},(i)} = f\left(\underset{y \in \{Y_{im}, \dots, Y_{(i+1)m-1}\}}{\arg\max} r(y)\right) \qquad (i = 1, \dots, k), \qquad (2)$$

$$Z_{m,n}^{\text{BoN + SC}} = \underset{z}{\arg\max} \sum_i \mathbb{I}\left[Z_m^{\text{Best},(i)} = z\right]. \qquad (3)$$

The main problem with BoN+SC is that it is too expensive. We would like to have a large value for $m$ to get the benefits offered by BoN. To have a fairly accurate estimation of $\pi_m$, we need a reasonably large value for $k$. Together, this requires a large budget $N \approx mk$.

The deficiency of BoN+SC comes from the fact that each sample $Y_i$ only contributes to generating one BoN output. To address this deficiency, we propose estimating $\pi_m$ not by generating independent samples from it, but by bootstrapping. To do that, we first note that the distribution $\pi_m$ of $Z_m^{\text{Best}}$ is a function of the unknown distribution $p_{\text{ref}}$. Bootstrapping suggests to estimate $\pi_m$ with the BoN's output distribution under a known approximation $\hat{p}_{\text{ref}} \approx p_{\text{ref}}$. The typical non-parametric approach is

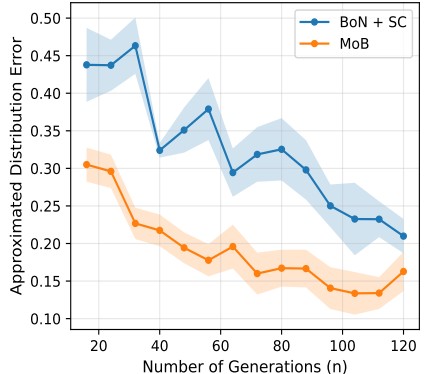
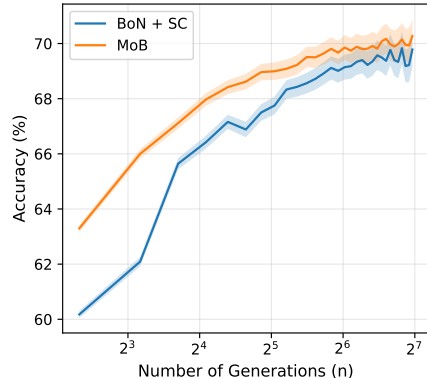

Figure 4: Comparison of MoB and BoN+SC using Qwen2.5-3B as the reference model and ArmoRM as the reward model. *(Left)* $\pi_m$ approximation error in $\ell_1$-norm for $m = 8$. *(Right)* Average accuracy on MMLU-Pro-Math dataset. Shaded area indicates the standard error.

to set $\hat{p}_{\text{ref}}$ to be the empirical distribution of the generated samples $\{Y_1, \ldots, Y_N\}$. Since $\hat{p}_{\text{ref}}$ is known, we can cheaply sample from it. For any arbitrarily large value $B$, we generate $B$ approximately sampled BoN outputs. We first create $B$ datasets of size $m$ from $\hat{p}_{\text{ref}}$. That is

$$D_i = \{\hat{Y}_{i,1}, \hat{Y}_{i,2}, \ldots, \hat{Y}_{i,m}\} \sim \hat{p}_{\text{ref}}, \qquad (i = 1, \ldots, B).$$

This is equivalent to sampling $m$ outputs from the original pool $\{Y_1, \ldots, Y_n\}$ with replacement. Then, similar to BoN+SC, we can run BoN on each dataset, and then pick the most common outcome. Formally,

$$\hat{Z}_m^{\text{Best},(i)} = f\left( \underset{y \in D_i}{\operatorname{argmax}} \, r(y) \right) \qquad (i = 1, \ldots, B), \qquad (4)$$

$$Z_{m,N}^{\text{MoB}} = \underset{z}{\operatorname{argmax}} \sum_{i=1}^{B} \mathbb{I}\left[ \hat{Z}_m^{\text{Best},(i)} = z \right]. \qquad (5)$$

This procedure is our MoB algorithm for a given $m$. We define $\hat{\pi}_{m,N}$ to be the (random) distribution of $\hat{Z}_m^{\text{Best},(1)}$ given $\{Y_i\}$ at hand. With sufficiently large $B$ (usually $B = 10,000$ is sufficient), the empirical distribution of $\{\hat{Z}_m^{\text{Best},(i)}\}$ will accurately estimate $\hat{\pi}_{m,N}$. With this approximation, we can write

$$Z_{m,N}^{\text{MoB}} \approx \underset{z}{\operatorname{argmax}} \, \hat{\pi}_{m,N}(z) \qquad (6)$$

Note that this is a light computation that can be carried out on the CPU. Therefore, we can freely choose a large $B$. In the supplementary material, we provide an even more efficient way of estimating $\hat{\pi}_{m,N}$ with $\mathcal{O}(N \log N)$ complexity that finds $Z_{m,N}^{\text{MoB}}$ directly and without creating $B$ datasets. It is worth mentioning that our use of bootstrap samples resembles bagging (Breiman, 1996) and subagging (Scornet et al., 2015), where a family of models is trained on the subsampled datasets and then aggregated.

In Figure 4, we compare MoB with BoN+SC in the same setup as Figure 3. In the left plot, we fix $m = 8$ and compare the algorithms' error on estimating $\pi_m$ for a range of values for $N$. We measure the distance between the two distributions according to the $\ell_1$-norm. As we can see, bootstrapping is consistently the superior approach for this approximation task and offers a more accurate estimation of $\pi_m$. In the right plot, we set $m = \lfloor \sqrt{N} \rfloor$ and compare the final accuracy of the algorithms. The choice of $m = \lfloor \sqrt{N} \rfloor$ ensures that $k \approx \sqrt{N}$ and will also increase as $N$ increases. We observe that the superior accuracy of bootstrapping in the estimation of $\pi_m$ translates to a better final accuracy of the algorithm, especially when the budget $N$ is more limited.

One might wonder if it is possible to choose $m$ to be much larger than what was possible in BoN+SC, potentially even $m = N$. There is no obvious limitation on the size of resampled datasets $D_i$, and nonetheless, most commonly in bootstrapping, the size of resampled datasets is equal to the original

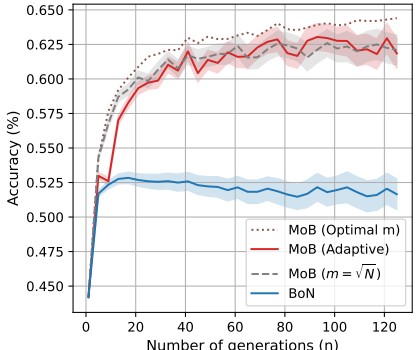 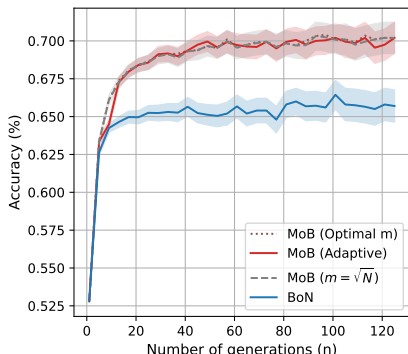

Figure 5: Comparing $m$ selection methods using ArmoRM reward model with MMLU-Pro-Math and Qwen2.5-3B *(Left)* and MATH500 and Llama3.1-8B *(Right)*. Shaded area indicates the standard error.

dataset. However, estimating the distribution of values related to the extremes of random samples is a classic example of failure for the conventional bootstrapping, see for example Athreya and Fukuchi (1994) and Efron and Tibshirani (1994, Section 7.4). Since BoN selects the output with the highest reward, it is affected by the same failure. To see this, note that the output with the highest reward appears in each dataset with the probability of $1 - (\frac{N-1}{N})^m$, and it will be chosen in any dataset in which it appears. Therefore, if $m = N$,

$$\mathbb{P}\Big(\hat{Z}_m^{\text{Best},(i)} = Z_N^{\text{Best}}\Big) \geq 1 - \big(\frac{N-1}{N}\big)^N \approx 1 - e^{-1} \approx 0.632.$$

This means that $\hat{\pi}_{N,N}$ will always incorrectly assign a probability of at least $0.632$ to the conventional BoN's answer.

Fortunately, using smaller resampled datasets, as we do in MoB, is one of the remedies for such failures of bootstrapping and is well-studied in the literature(Athreya and Fukuchi, 1994; Bickel et al., 2011). This approach is referred to as $m$-out-of-$n$ bootstrapping. We show that under the usual conditions of $m$-out-of-$n$ bootstrapping and mild assumptions on the tail of reward distributions, our use of bootstrapping to estimate $\pi_m$ is a valid one. Similar to the typical guarantees for bootstrap estimations, we show that our bootstrap estimation is indeed consistent.

**Theorem 1.** *Under mild assumptions on the distribution of rewards, if there are finite possible values for $Z$ and as $N \to \infty$, we have $m \to \infty$ and $m/N \to 0$, then the estimated $\hat{\pi}_{m,N}$ will converge to the true distribution $\pi_m$. That is, for any $\epsilon > 0$,*

$$\lim_{N \to \infty} \mathbb{P}\big(\|\hat{\pi}_{m,N} - \pi_m\|_1 \geq \epsilon\big) = 0.$$

We defer the exact technical statement and proof to the supplementary material. Theorem 1 shows that the estimated distribution $\hat{\pi}_{m,N}$ will match the true BoN output distribution $\pi_m$. It means that MoB with bootstrapped distribution in (6) will reach the same accuracy as its oracle version in (1), but with a larger required budget due to $m < N$. To achieve this, it suffices to pick $m$ such that the condition of Theorem 1 holds, which is possible by simply using a fixed schedule of the form $m(n) = n^\alpha$ for some $0 < \alpha < 1$. In the next section, we will discuss the choice of $m$ in more detail and provide a procedure to choose $m$ automatically.

### 4.3 Adaptive Subsample Size $m$

The choice of $m$ imposes a trade-off. A larger value of $m$ means that we are running BoN with a larger number of samples. Since we expect the success probability of BoN to increase with more samples, this means that the mode of $\pi_m$ will be more likely to be correct. On the other hand, as $m$ becomes larger and closer to $n$, our estimate $\hat{\pi}_{m,N}$ of $\pi_m$ becomes more inaccurate. As we saw in Section 4.2, bootstrapping might fail to provide a consistent estimate if $m = N$.

Ideally, we would like to find an $m$ such that our final answer $Z_{m,N}^{\text{MoB}}$ based on the estimated distribution as in (6) becomes closest to the Oracle MoB (1) of Section 4.1. The natural approach for this goal is

to find the value of $m$ that minimizes the distance between $\hat{\pi}_{m,N}$ and $\pi_N$, that is

$$M_N^* = \underset{m}{\operatorname{argmin}} \|\hat{\pi}_{m,N} - \pi_N\|_1. \tag{7}$$

This minimization problem automatically captures both aspects of the trade-off. Large values of $m$ make $\pi_m$, which is approximated by $\hat{\pi}_{m,N}$ closer to $\pi_N$, but at the same time if $m$ is too large, the error of this approximation becomes too large and increases the objective $\|\hat{\pi}_{m,N} - \pi_N\|_1$.

Unfortunately, the distribution $\pi_N$ in the objective of (7) is unknown, and therefore cannot be used in practice. The theoretical results by Götze and Račkauskas (2001) show that if $Z$ only takes two possible values and under some other technical conditions, the distance $\|\hat{\pi}_{m,N} - \hat{\pi}_{m/2,N}\|_1$ is proportional to the one in (7)[3]:

$$\|\hat{\pi}_{m,N} - \hat{\pi}_{m/2,N}\|_1 \propto \|\hat{\pi}_{m,N} - \pi_N\|_1.$$

Inspired by this result, Bickel and Sakov (2008) provides some optimality results for choosing $m$ by minimizing the more general loss $\|\hat{\pi}_{m,N} - \hat{\pi}_{qm,N}\|_1$ for some $0 < q < 1$ instead of just $q = 0.5$ considered by Götze and Račkauskas (2001).

Based on the findings of Bickel and Sakov (2008), we propose using the following approach to pick $m$. We first consider the candidates of the form $\lfloor q^j N \rfloor$ and pick the value among them that minimizes $\|\hat{\pi}_{m,N} - \hat{\pi}_{qm,N}\|_1$.

$$m_j = \lfloor q^j N \rfloor \qquad (j = 0, 1, 2, \ldots),$$
$$\hat{M}_N^* = \underset{m=m_j}{\operatorname{argmin}} \|\hat{\pi}_{m_j,N} - \hat{\pi}_{m_{j-1},N}\|_1.$$

Note that this involves calculating $\hat{\pi}_{m,N}$ for all values of $m_j$. These will be just $\mathcal{O}(\log N)$ distributions and computationally cheap. Finally, output selected by MoB with adaptive $m$ is $Z_N^{\mathrm{MoB}} = Z_{\hat{M}_N^*,N}^{\mathrm{MoB}}$.

The choice of $q$ has been observed not to be critical in most applications. Bickel and Sakov (2008) observes no significant difference among $q = 0.75, 0.65, 0.6, 0.5$. In our experiments, we fix $q = 0.75$. In Figure 5, we evaluate the efficiency of this procedure to select $m$. For each $N$, we measure the highest accuracy achieved by MoB when choosing $m$ from $\{N^\alpha\}$ for $\alpha \in [0.1, 0.9]$. We plot the accuracy of our adaptive $m$ as well as $m = \sqrt{N}$ approach against this optimal performance for two different settings. These figures show both that adaptive $m$ and the simple $m = \sqrt{N}$ achieve performance close to the optimal $m$ variant. This indicates MoB's performance is not sensitive to the choice of $m$ and both the adaptive and simple square root choices achieve a near-optimal performance without hyperparameter tuning. We repeat this comparison in more settings in Appendix E.1.

## 5 Intuitions and Conditions for Improvement

We now investigate why and when MoB outperforms BoN. To this end, we decompose the algorithmic changes from BoN to MoB into two steps, allowing us to study their individual impact more easily. MoB is the result of applying these two changes to BoN:

1. **Change from Best-of-$N$ to Best-of-$m$.** To be able to approximate BoN's output distribution, MoB is forced to work with $\pi_m$, the output distribution of Best-of-$m$, for some $m < N$.

2. **Change from Best-of-$m$ to MoB.** The output of Best-of-$m$ is a sample from its output distribution $\pi_m$. On the other hand, MoB estimates $\pi_m$ and selects its mode.

Together, the effects of these two steps determine whether MoB outperforms BoN. The impact of each step varies by question and depends on the base model's generation distribution and the reward model's reward distribution for that question. These distributions—especially the reward distribution—can be complex. To enable an intuitive analysis, we measure two metrics for each question: the base model's success probability and the reward model's accuracy, defined as the fraction of incorrect–correct output pairs in which the correct output receives a higher reward. We

---

[3]This is a rough interpretation of the results by Götze and Račkauskas (2001), where the ratio of the two losses is studied. We refer the reader to the original paper for more details.

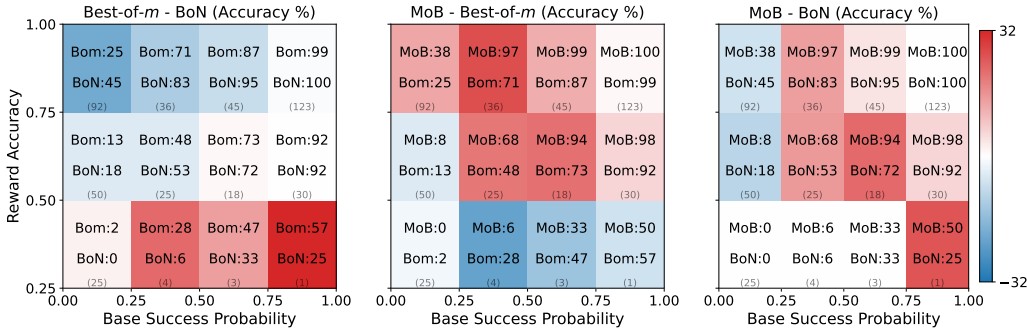

Figure 6: Comparison of Best-of-$m$ ($m$ chosen adaptively) vs. Best-of-$N$ *(Left)*, MoB vs. Best-of-$m$ *(Middle)*, and MoB vs. BoN *(Right)*. Numbers in parentheses are the size of each group.

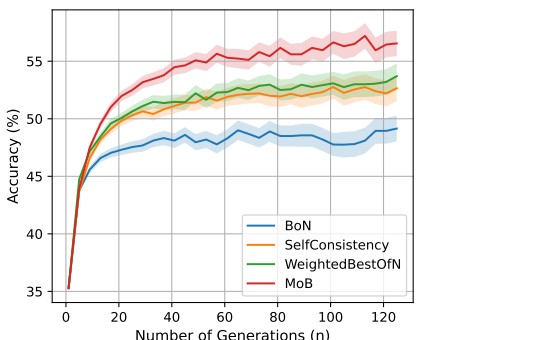 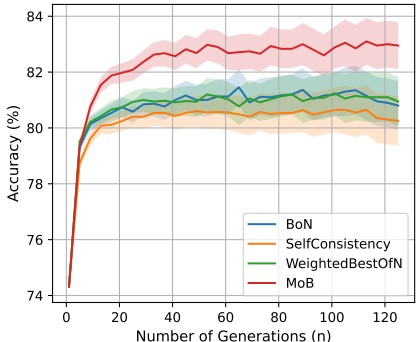

Figure 7: Accuracy comparison on different datasets using base model Qwen2.5-3B and GRM reward model. Standard deviation is shown as the shaded area. *(Left)*: MMLU-Pro-Chem *(Right)*: GSM8k

then categorize questions into 12 groups according to the value of these two metrics and analyze the effects of our changes in each group.

In Figure 6, we compare the accuracy of Best-of-$N$, Best-of-$m$, and MoB across the groups for MMLU-Pro-Math benchmark with Gemma2-9B base model, ArmoRM reward model, and $N = 128$. The left plot compares Best-of-$N$ with Best-of-$m$ to show the impact of the first step. In questions with accurate rewards but weak base model performance, BoN benefits the most with more outputs. Therefore, we observe that using $m$ outputs instead of $N$ has the most negative effect on the performance. The performance of Best-of-$m$ is compared to MoB in the middle plot of Figure 6 to measure the impact of the second step. In questions where the mode is correct, MoB will outperform Best-of-$m$ by picking the correct answer with high probability, even if its chance of selection by Best-of-$m$ is low. On the other hand, if the mode is incorrect, MoB will be wrong with high probability, but Best-of-$m$ can still solve the problem by chance. The effect of choosing the mode instead of sampling on the performance depends on the relative number of these two kinds of questions. We expect the mode to be correct more often in groups with accurate enough rewards and base model generations. This is verified by our observation where we see the highest improvement by MoB over Best-of-$m$ for these questions.

Lastly, the right plot in Figure 6 shows the combined effect of the two changes and compares MoB with BoN in each group. MoB improves upon BoN the most in questions where the reward and base models are good enough to make the mode correct, but are not good enough to achieve near perfect accuracy. In Appendix C, we study the success probability of MoB and BoN in a synthetic setup with $N = \infty$ and make similar observations.

# 6 Experiments

We conducted a series of experiments to compare the performance of our proposed method against other well-known sample-and-marginalize approaches across a range of datasets, generative models, and reward models. The datasets include MATH500 (Lightman et al., 2023), GSM8K (Cobbe et al., 2021b), MMLU-Pro (Wang et al., 2024b) questions in math (MMLU-Pro-Math) and chemistry (MMLU-Pro-Chem), and CommonSenseQA (Talmor et al., 2018). We have experimented with three different generative models from different families and different sizes: Qwen2.5-3B-Instruct (Qwen Team, 2024), Llama-3.1-8B-Instruct (Grattafiori et al., 2024), and Gemma-2-9B-it (Team et al., 2024). For reward models, we used two widely adopted ORMs: ArmoRM (Wang et al., 2024a) and GRM (Yang et al., 2024), with 8B and 3B parameters, respectively. These choices result in thirty diverse experimental setups that rigorously evaluate our method's performance.

Figure 7 presents the accuracy of different methods on GSM8K and MMLU-Pro-Chem across varying values of $N$. Our method consistently outperforms the baselines, showing clear improvements even at smaller $N$ values. Table 1 presents the accuracy and its standard error for MoB with adaptive $m$ and $m = \sqrt{N}$ alongside SC, BoN, and WBoN for $N = 128$ across all benchmarks, using Qwen and GRM models. Algorithms with statistically insignificant difference to the best algorithm according to a paired one-sided t-test (p-value $> 0.05$) are also shown in bold. In Table 2, we report the accuracy on MATH500 for all base and reward model combinations. This table also includes a row showing the performance improvement of our method over BoN. As shown in both tables, MoB consistently outperforms BoN in every setting. These results show the potential of MoB as a strong replacement of BoN in tasks with discrete final answers. Complete results for all thirty experiment configurations are provided in Appendix E.2.

Table 1: Results for Qwen2.5-3B and GRM as base and reward models ($N = 128$).

|  | MATH500 | MMLU-Pro-Math | MMLU-Pro-Chem | GSM8K | CSQA |
|---|---|---|---|---|---|
| BoN | 63.95±1.07 | 66.10±1.06 | 49.00±1.12 | 80.95±0.88 | **77.70±0.93** |
| SC | 66.40±1.06 | 65.60±1.06 | 52.50±1.12 | 80.40±0.89 | 76.20±0.95 |
| WBoN | 67.45±1.05 | 64.35±1.07 | 53.10±1.12 | 81.25±0.87 | 54.90±1.11 |
| MoB-Adaptive (Ours) | **69.95±1.03** | 69.30±1.03 | **56.45±1.11** | **82.85±0.84** | 77.40±0.94 |
| MoB-Poly (Ours) | **69.45±1.03** | **70.15±1.02** | **56.30±1.11** | **83.10±0.84** | 77.45±0.93 |
| ↑MoB over BoN | 6.00±0.78 | 3.20±0.79 | 7.45±0.94 | 1.90±0.51 | -0.30±0.52 |

Table 2: Results on MATH500 across all base and reward models ($N = 128$).

|  | **ArmoRM** | | | **GRM** | | |
|---|---|---|---|---|---|---|
|  | Llama3.1-8B | Gemma2-9B | Qwen2.5-3B | Llama3.1-8B | Gemma2-9B | Qwen2.5-3B |
| BoN | 51.55±1.12 | 52.20±1.12 | 60.60±1.09 | 56.65±1.11 | 54.95±1.11 | 63.95±1.07 |
| SC | 60.65±1.09 | 52.90±1.12 | 66.40±1.06 | 60.65±1.09 | 52.90±1.12 | 66.40±1.06 |
| WBoN | **62.90±1.08** | 53.85±1.11 | 67.10±1.05 | **63.55±1.08** | 56.15±1.11 | 67.45±1.05 |
| MoB-Adaptive (Ours) | **62.90±1.08** | 56.15±1.11 | **68.50±1.04** | **64.30±1.07** | 57.45±1.11 | **69.95±1.03** |
| MoB-Poly (Ours) | **62.40±1.08** | **57.05±1.11** | 67.85±1.04 | **64.00±1.07** | **58.10±1.10** | **69.45±1.03** |
| ↑MoB over BoN | 11.35±0.86 | 3.95±0.68 | 7.90±0.78 | 7.65±0.80 | 2.50±0.64 | 6.00±0.78 |

# 7 Conclusion and Future Work

In this paper we highlighted that with imperfect rewards, BoN's chosen answer can be highly stochastic and fail to pick the correct answer reliably. To address this, we introduced Majority-of-the-Bests (MoB), which estimates BoN's output distribution via bootstrapping and chooses the most probable outcome. MoB achieves superior performance compared to other selection algorithms such as, BoN, Self-consistency, and Weighted BoN outperforming them in most of our 30 experimental setups. Our method is scalable, requires no hyperparameter tuning, and adds only negligible CPU computational overhead. MoB can serve as a strong alternative to BoN and SC in problems with discrete final answers. Looking forward, we believe MoB's selection signal could enable early stopping in parallel LLM generation, or be applied more broadly in any framework that relies on sampling from an LLM. However, MoB is limited to settings where the task requires producing a final answer, and like all sampling-based methods, it incurs higher inference costs compared to zero-shot approaches.

## Acknowledgments and Disclosure of Funding

We thank the anonymous reviewers who provided valuable feedback that led to significant improvements to the paper. AMF acknowledges the funding from the Natural Sciences and Engineering Research Council of Canada (NSERC) through the Discovery Grant program (2021-03701).

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

## List of Appendices

We provide a brief description of the material in the appendix of the paper.

## A  Theoretical Results

In this section, we provide the formal theoretical results and the proof of Theorem 1. To do so, we first need to show the convergence of BoN's output distribution, which is done in Section A.1 and Theorem 2. We prove Theorem 1 in Section A.2.

### A.1  Asymptotic Behavior of BoN's Output Distribution

**Theorem 2.** *For final answer $z$ such that $\pi_{ref}(z) \in (0, 1)$, let $F_0$ and $F_1$ represent cumulative distribution functions (CDFs) of the conditional distributions $\mathbb{P}(r(Y)|f(Y) = z)$ and $\mathbb{P}(r(Y)|f(Y) \neq z)$, respectively. Define $x_0$ and $x_1$ to be their right endpoints,*

$$x_0 \triangleq \sup\{x \in \mathbb{R} : F_0(x) < 1\}, \quad x_1 \triangleq \sup\{x \in \mathbb{R} : F_1(x) < 1\}.$$

*As $N \to \infty$,*

*(i) if $x_0 < x_1$, we have $\pi_N(z) \to 0$.*

*(ii) if $x_0 > x_1$, we have $\pi_N(z) \to 1$.*

*(iii) if $x_0 = x_1 = x^*$, $F_0$ and $F_1$ are continuous and strictly increasing, and for some $c \in [0, \infty]$,*

$$\lim_{x \uparrow x^*} \frac{1 - F_0(x)}{1 - F_1(x)} = c, \tag{1}$$

*then we have,*

$$\pi_N(z) \to \frac{c \cdot \pi_{ref}(z)}{1 + (c - 1) \cdot \pi_{ref}(z)}.$$

*Proof.* We first define some random variables to better express $\pi_N(z)$. Assume we use $F_0$ and $F_1$ to generate i.i.d. samples $R_1^0, R_2^0, \ldots \overset{\text{i.i.d.}}{\sim} F_0$ and $R_1^1, R_2^1, \ldots \overset{\text{i.i.d.}}{\sim} F_1$. For $n \geq 1$, let $S_n^0$ and $S_n^1$ be the maximum of the first $n$ samples from $F_0$ and $F_1$, that is,

$$S_n^0 \triangleq \max_{i=1,\ldots,n} R_i^0 \quad , \quad S_n^1 \triangleq \max_{i=1,\ldots,n} R_i^1.$$

Also, for outputs $Y_1, \ldots, Y_N$, let $Z_i = f(Y_i)$, $N^0$ be the number of outputs that reach the final answer $z$, and $N^1 = N - N^0$ be the number of outputs that do not reach the final answer $z$.

We can express $\pi_N(z)$ as

$$\pi_N(z) = \sum_{z_{1:N}} \mathbb{P}\big(Z_N^{\text{Best}} = z | Z_{1:N} = z_{1:N}\big) \cdot \mathbb{P}(Z_{1:N} = z_{1:N})$$

$$= \sum_{z_{1:N}} \mathbb{P}\bigg(\max_{z_i = z} r(Y_i) > \max_{z_i \neq z} r(Y_i) | Z_{1:N} = z_{1:N}\bigg) \cdot \mathbb{P}(Z_{1:N} = z_{1:N}). \tag{2}$$

Now, note that due $Y_1, \ldots, Y_N$ being i.i.d., we have

$$\mathbb{P}(r(Y_1), \ldots, r(Y_N) | Z_{1:N} = z_{1:N}) = \prod_i \mathbb{P}(r(Y_i) | Z_i = z_i).$$

By definition of $R_i^0$ and $R_i^1$, we can therefore write (2) as

$$\pi_N(z) = \sum_{z_{1:N}} \mathbb{P}(S_{N^0}^0 > S_{N^1}^1 | Z_{1:N} = z_{1:N}) \cdot \mathbb{P}(Z_{1:N} = z_{1:N}) = \mathbb{P}(S_{N^0}^0 > S_{N^1}^1).$$

For simplicity, we define $S^1 \triangleq S_{N^1}^1$ and $S^0 \triangleq S_{N^0}^0$. Now, we can express $\pi_N(z)$ as

$$\pi_N(z) = \mathbb{P}(S^0 > S^1).$$

Note that $S^0 \xrightarrow{d} x_0$ and $S^1 \xrightarrow{d} x_1$, which leads to the statement for cases *(i)* and *(ii)*. We focus on case *(iii)*. Let $\bar{F}_0(x) \triangleq 1 - F_0(x)$ and $\bar{F}_1(x) \triangleq 1 - F_1(x)$ be the complementary CDFs of $F_0$ and $F_1$, respectively. To quantify $\mathbb{P}(S^0 > S^1)$, we note that $\bar{F}_1$ is strictly decreasing in a neighborhood of $S^1$. Thus,

$$\lim_{N \to \infty} \pi_N(z) = \lim_{N \to \infty} \mathbb{P}(S^0 > S^1) = \lim_{N \to \infty} \mathbb{P}(N \bar{F}_1(S^0) < N \bar{F}_1(S^1)). \tag{3}$$

Therefore, we turn to study the joint distribution of $(N \bar{F}_1(S^0), N \bar{F}_1(S^1))$ as $N \to \infty$. This will be achieved by quantifying the distribution of $(n_0 \bar{F}_1(S_{n_0}^0), n_1 \bar{F}_1(S_{n_1}^1))$ as $n_0, n_1 \to \infty$ and relating it to the distribution of $(N \bar{F}_1(S^0), N \bar{F}_1(S^1))$.

Since $F_1$ is continuous, $F_1(R_i^1) \sim U[0, 1]$ is uniformly distributed for any $i$. Define $U_i = \bar{F}_1(R_i^1) \sim U[0, 1]$. It is well known that

$$n_1 \min_{i=1, \ldots, n_1} U_i \xrightarrow{d} \mathrm{Exp}(1) \qquad\qquad (n_1 \to \infty),$$

which due to $\min_i \bar{F}_1(R_i^1) = \bar{F}_1(S_{n_1}^1)$, translates to

$$n_1 \bar{F}_1(S_{n_1}^1) \xrightarrow{d} \mathrm{Exp}(1) \qquad\qquad (n_1 \to \infty). \tag{4}$$

Similarly, we can show that $n_0 \bar{F}_0(S_{n_0}^0) \xrightarrow{d} \mathrm{Exp}(1)$ as $n_0 \to \infty$. However, our goal is to analyze the distribution of $n_0 \bar{F}_1(S_{n_0}^0)$. To do so, we use the tail-equivalence condition (1). We note that $S_{n_0}^0 \xrightarrow{d} x^*$, therefore, $\bar{F}_0(S_{n_0}^0) / \bar{F}_1(S_{n_0}^0) \xrightarrow{d} c$ as $n_0 \to \infty$. Together, we get

$$n_0 \bar{F}_1(S_{n_0}^0) = \frac{n_0 \bar{F}_0(S_{n_0}^0)}{\bar{F}_0(S_{n_0}^0) / \bar{F}_1(S_{n_0}^0)} \xrightarrow{d} \frac{\mathrm{Exp}(1)}{c} \qquad\qquad (n_0 \to \infty). \tag{5}$$

Due to the independence of $S_{n_1}^1$ and $S_{n_0}^0$, we can combine (4) and (5) to get

$$(n_0 \bar{F}_1(S_{n_0}^0), n_1 \bar{F}_1(S_{n_1}^1)) \xrightarrow{d} (E/c, F) \qquad\qquad (n_0, n_1 \to \infty),$$

where $E, F \overset{\text{i.i.d.}}{\sim} \mathrm{Exp}(1)$. As $N \to \infty$, we have $N^0, N^1 \xrightarrow{P} \infty$, therefore,

$$(N^0 \bar{F}_1(S_{N^0}^0), N^1 \bar{F}_1(S_{N^1}^1)) \xrightarrow{d} (E/c, F) \qquad\qquad (N \to \infty).$$

Finally, we use the fact that $N^0/N \xrightarrow{d} \pi_{\mathrm{ref}}(z)$ and $N^1/N \xrightarrow{d} 1 - \pi_{\mathrm{ref}}(z)$ to get

$$(N \bar{F}_1(S^0), N \bar{F}_1(S^1)) = \left( \frac{N^0 \bar{F}_1(S_{N^0}^0)}{N^0/N}, \frac{N^1 \bar{F}_1(S_{N^1}^1)}{N^1/N} \right) \xrightarrow{d} \left( \frac{E}{c \cdot \pi_{\mathrm{ref}}(z)}, \frac{F}{1 - \pi_{\mathrm{ref}}(z)} \right). \tag{6}$$

Combined with (3), we conclude that

$$\lim_{N \to \infty} \pi_N(z) = \mathbb{P}\left( \frac{E}{c \cdot \pi_{\mathrm{ref}}(z)} < \frac{F}{1 - \pi_{\mathrm{ref}}(z)} \right) = \frac{c \pi_{\mathrm{ref}}(z)}{1 - \pi_{\mathrm{ref}}(z) + c \pi_{\mathrm{ref}}(z)}.$$

$\square$

## A.2 Proof of Theorem 1

We restate Theorem 1 with the assumptions not included in the main text.

**Theorem 3.** *Assume that there are finite possible values for $Z$ and for every possible final answer $z$, the conditions of Theorem 2 for one the cases hold. If as $N \to \infty$, we have $m \to \infty$ and $m/N \to 0$, then for any $\epsilon > 0$, the estimated $\hat{\pi}_{m,N}$ will converge to the true distribution $\pi_m$. That is,*

$$\lim_{n \to \infty} \mathbb{P}\big(\|\hat{\pi}_{m,N} - \pi_m\|_1 \geq \epsilon\big) = 0.$$

*Proof.* Since there are finite possible values for $Z$, it suffices to show the convergence in estimated probability of each possible final answer $z$. We show that for any $z$, and $\epsilon > 0$, we have

$$\lim_{N \to \infty} \mathbb{P}(|\hat{\pi}_{m,N}(z) - \pi_m(z)| \geq \epsilon) = 0. \tag{7}$$

We use the result by Bickel et al. (2011, Equation 3.14) to show this claim. To do so, we first frame our problem in their notation. For $1 \leq i \leq N$, let $Z_i \triangleq f(Y_i)$ be (the one-hot encoding of) the final answer reached by $Y_i$, and $R_i \triangleq r(Y_i)$ be the numerical reward of $Y_i$. We define

$$X_i \triangleq (Z_i, R_i).$$

We define the bootstrap statistic of $X_1, \ldots, X_m$ as

$$T_m = \mathbb{I}\big[Z_m^{\text{Best}} = z\big] + \frac{D}{4} \sim L_m,$$

where $\mathbb{I}[\cdot]$ is the indicator function, $D \sim \text{Bernoulli}(0.5)$ is an independent Bernoulli random variable, and $L_m$ is defined to be the distribution of $T_m$. Basically, $T_m$ is the indicator of $z$ being selected by BoN, plus a small random noise to ensure the non-degeneracy condition as $m \to \infty$. We define the function $h(t) = \mathbb{I}[t > 0.5]$, so that the parameter of interest $\theta_m$ becomes

$$\theta_m \triangleq \mathbb{E}h(T_m) = \pi_m(z),$$

as intended. Lastly, one can verify that since $T_m$ is invariant of repetitions and permutations of its inputs $X_1, \ldots, X_m$, in our case, we have for any $0 < x < 1$,

$$\delta_m(x) \triangleq \big|\pi_{\lfloor mx \rfloor}(z) - \pi_m(z)\big|.$$

We now show the conditions of Bickel et al. (2011, Theorem 2). First, we need to show that $L_m$, the distribution of $T_m$, is convergent. According to Theorem 2, we have

$$\lim_{m \to \infty} \pi_m(z) \triangleq \pi_\infty(z)$$

for some $\pi_\infty(z) \in [0, 1]$. Therefore, as $m \to \infty$, we have

$$L_m \xrightarrow{d} \text{Bernoulli}(\pi_\infty(z)) + \frac{\text{Bernoulli}(0.5)}{4}.$$

For condition Bickel et al. (2011, Equation 3.11) we need to show that for any $M < \infty$, we have

$$\delta_m(1 - xm^{-1/2}) \to 0$$

uniformly for all $0 < x < M$. By definition, it suffices to show that for any $0 < x < M$, we have

$$\big|\pi_{\lfloor m - x\sqrt{m} \rfloor}(z) - \pi_m(z)\big| \to 0.$$

This follows from the fact that $\pi_m(z)$ is convergent to $\pi_\infty(z)$. For any $\varepsilon > 0$, pick $M_0$ such that for any $m_0 \geq M_0$, we have

$$\big|\pi_{m_0}(z) - \pi_\infty(z)\big| < \frac{\varepsilon}{2},$$

and $M_1$ such that for any $M_1 - M\sqrt{M_1} \geq M_0$. Then for any $m \geq M_1$, we have

$$\big|\pi_{\lfloor m - x\sqrt{m} \rfloor}(z) - \pi_\infty(z)\big| < \varepsilon/2 \quad \text{and} \quad \big|\pi_m(z) - \pi_\infty(z)\big| < \varepsilon/2.$$

Together, we have

$$\big|\pi_{\lfloor m - x\sqrt{m} \rfloor}(z) - \pi_m(z)\big| < \varepsilon$$

and achieve the uniform convergence condition.

Finally, note that our statistic $T_m$ is not dependent on the sampling distribution $p_{\text{ref}}$ and Bickel et al. (2011, Equation 3.13) is satisfied. □

## B   Closed-Form Calculation of Bootstrapped BoN's Output Distribution

In Section 4.2, we proposed approximating $\hat{\pi}_{m,N}$ by running BoN on a large number $B$ of subsets of size $m$ sampled with replacement from the $N$ generated outputs. In practice, $B = 10,000$ is commonly considered sufficient. This calculation is negligible compared to the generation of outputs from the LLM and can be carried out on a CPU. Nonetheless, we here show that it can also be done in $\mathcal{O}(N \log N)$.

Define $R_i = r(Y_i)$ for $1 \leq i \leq N$, and let $i_1, i_2, \ldots, i_N$ be such that

$$R_{i_1} < R_{i_2} < \ldots < R_{i_N}.$$

For simplicity, we assume no ties occur among the rewards. The key insight is that for any $1 \leq k \leq N$, the probability of $Y_{i_k}$ being selected in a randomly sampled subset of $m$ outputs can be calculated in closed-form. We note that $Y_{i_k}$ is selected if the subset only includes outputs among $Y_{i_1}, \ldots, Y_{i_k}$, but is not limited to $Y_{i_1}, \ldots, Y_{i_{k-1}}$ (and therefore contains $Y_{i_k}$). We get

$$\mathbb{P}(Y_{i_k} \text{ is the output of BoN on a resampled subset}) = \left(\frac{k}{N}\right)^m - \left(\frac{k-1}{N}\right)^m.$$

Thus, for any final answer $z$, the probability of it being selected in a subset is

$$\hat{\pi}_{m,N}(z) = \sum_{k:Z_{i_k}=z} \left(\frac{k}{N}\right)^m - \left(\frac{k-1}{N}\right)^m.$$

This procedure only requires sorting the outputs according to their rewards and therefore has complexity of $\mathcal{O}(N \log N)$.

## C   Effect of Reward Noise and Base Model's Success Probability

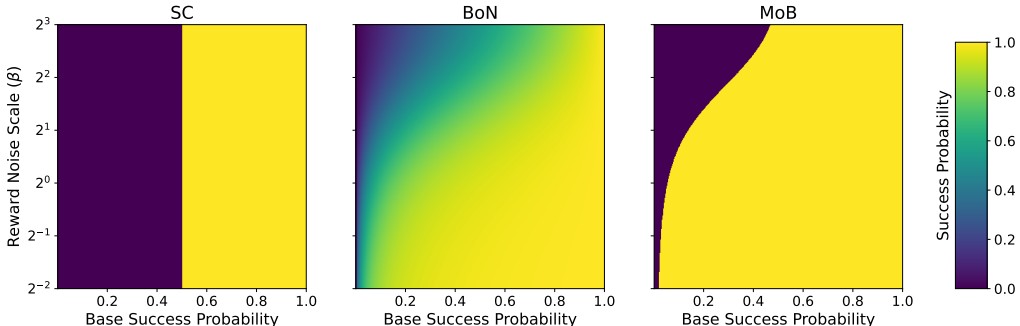

Figure 8: Success probability of SC, BoN, and MoB with infinite budget ($N = \infty$) for different values of the base model's success probability and reward noise.

In this section, we investigate the effect of the base model and reward noise on the success probability of SC, BoN, and MoB. We consider a synthetic setup for a TRUE/FALSE question, where the correct answer is TRUE. Let $p$ be the success probability of the base model, which is the probability that the base model generates a solution reaching the correct final answer.

Assume $r^{\text{oracle}}$ is an oracle reward model that always assigns the reward of 1 to solutions that reach the correct answer, and 0 otherwise:

$$r^{\text{oracle}}(Y) = \begin{cases} 1, & \text{if } f(Y) = \text{TRUE}, \\ 0, & \text{if } f(Y) = \text{FALSE}. \end{cases}$$

To investigate the effect of an imperfect reward model, we consider a noisy reward model $r^{\text{noisy}}$ that is equal to the oracle reward plus an exponentially distributed noise:

$$r^{\text{noisy}}(Y) = r^{\text{oracle}}(Y) + \text{Exp}(1/\beta).$$

The parameter $\beta$ controls the noise level, where a larger $\beta$ indicates a noisier reward model. To see this, note that the expected value and the standard deviation of the noise are equal to $\beta$. If $\beta$ is large, the noise will dominate the signal from the oracle reward, and the noisy reward model will be less informative.

We visualize the success probability of SC, BoN, and MoB with infinite budget $N = \infty$ in Figure 8. SC's success probability, as shown in the left plot of Figure 8, is independent of the reward noise. It is either equal to 1 when $p > 0.5$ (the correct answer is the most probable answer), or equal to 0 otherwise. For BoN, consider two extreme cases for the reward noise. When the reward model is perfect ($\beta$ small), BoN's success probability is 1 regardless of the base model's success probability. This is shown in the bottom edge of the middle plot in Figure 8. In this case, BoN is preferable over SC. On the other hand, when the reward model is completely uninformative ($\beta$ large), BoN's success probability is equal to the base model's success probability. This is shown in the top edge of the middle plot in Figure 8. MoB's success probability is equal to 1 if BoN's success probability is at least 0.5, as shown in the right plot of Figure 8. We see that MoB shows a similar behavior to SC when the reward model is uninformative, and when the reward model is perfect, MoB behaves like BoN.

In this setup, we can study the success probability of BoN and MoB with an infinite budget $N = \infty$ theoretically. BoN's success probability depends on the reward's noise level. It can be calculated from Theorem 2 as

$$\text{BoN success probability with infinite budget} = \frac{e^{1/\beta}p}{1 - p + e^{1/\beta}p}.$$

Note that if the reward model is perfect ($\beta = 0$), both the numerator and denominator go to infinity, and we reach the success probability of 1. With $\beta = \infty$, the noise becomes dominant, and BoN's success probability remains equal to the base model $p$ even with infinite budget. Due to Theorem 1, MoB solves the problem if the correct answer is BoN's most probable outcome. Therefore,

$$\text{MoB success probability with infinite budget} = \begin{cases} 1, & \text{if } \frac{e^{1/\beta}p}{1-p+e^{1/\beta}p} > 0.5, \\ 0, & \text{otherwise.} \end{cases}$$

This is favorable over BoN in scenarios where BoN still prefers the correct answer, as it can find the correct answer reliably without randomness.

## D  Implementation and Experiment Details

In this section, we provide more details on how the experiments in the paper are conducted.

### D.1  Evaluation Experiments

**Benchmarks.** We run our experiments on five popular benchmarks. MATH500, first introduced by Lightman et al. (2023), is a randomly sampled subset of 500 math questions with short final answers from the MATH dataset (Hendrycks et al., 2021). We use the math and chemistry questions from the MMLU-Pro benchmark (Wang et al., 2024b), which includes multiple-choice questions on a variety of topics. We also run our experiments on GSM8K (Cobbe et al., 2021a) that contains grade school math questions in short final answer format. Lastly, we use the CommonsenseQA benchmark (Talmor et al., 2019) that tests the model's commonsense reasoning through multiple-choice questions. For all benchmarks, we randomly select 500 questions for our experiments.

**Base and Reward Models.** We have used the models `Qwen/Qwen2.5-3B-Instruct`, `google/gemma-2-9b-it`, and `meta-llama/Llama-3.1-8B-Instruct` from Huggingface as base generative models. Our reward models are `Ray2333/GRM-Llama3.2-3B-rewardmodel-ft` and `RLHFlow/ArmoRM-Llama3-8B-v0.1`, which are the best performing 3B and 8B reward models according to Rewardbench (Lambert et al., 2024) in reasoning tasks.

**Implementation Details.** In the implementation of MoB, we always use the closed-form calculation of $\hat{\pi}_{m,N}$ discussed in Appendix B to efficiently perform the bootstrap estimate. Therefore, in the actual implementation, there is no parameter $B$ and we effectively operate as if $B = \infty$ was chosen.

We use Huggingface's Python library for all the output generations. The generation was carried on H100 GPUs. The compute cost was not tracked, but we estimate it to be on the order of a few thousand GPU-hours. We always use temperature 1 for inference and no extra modification of the next-token sampling procedure. The final answer extraction and evaluation are calculated using the Language Model Evaluation Harness (Gao et al., 2024). For each question, we generate 512 outputs and for each budget size $N$, we run each algorithm $\lfloor 512/N \rfloor$ times. Reported standard errors for the accuracies are calculated with the assumption of normal errors and from the standard deviation of $500 \times \lfloor 512/N \rfloor$ independent runs of the algorithm (500 is the dataset size across all benchmarks). The numbers reported as the improvement of MoB over BoN are based on the adaptive MoB, and its standard error is calculated from the standard deviation of paired differences of the algorithms score in $500 \times \lfloor 512/N \rfloor$ runs. We use the Scipy library (Virtanen et al., 2020) in python to conduct one-sided paired t-test to decide statistical significance of the difference between the best performing algorithm with another algorithm in our tables. Algorithms with insignificant (p-value $> 0.05$) difference are also shown in bold.

For GSM8K, we use a 5-shot prompt. For MATH and MMLU-Pro questions, we use the zero-shot chain-of-thought prompting used in the official Llama3.1 models evaluation (Grattafiori et al., 2024) on MATH (Hendrycks et al., 2021). This prompt and the prompt used for CommonsenseQA are given in the following.

> **Prompt for MATH and MMLU-Pro**
>
> ```
> Solve the following <topic> problem efficiently and clearly:
> - For simple problems (2 steps or fewer):  Provide a concise solution
> with minimal explanation.
> - For complex problems (3 steps or more):  Use this step-by-step format:
> ## Step 1:  [Concise description] [Brief explanation and calculations]
> ## Step 2:  [Concise description] [Brief explanation and calculations]
> ...
> Regardless of the approach, always conclude with:
> Therefore, the final answer is:  $\\boxed{answer}$. I hope it is
> correct.
> Where [answer] is just the final number or expression that solves the
> problem.
> Problem:  <problem from dataset>
> ```

> **Prompt for CommonsenseQA**
>
> ```
> Use commonsense to solve the following multiple choice question.  First
> explain your solution and then give the final answer.  Always finish
> your answer with "the answer is (X)" where X is the correct letter
> choice.  Question::  <problem from dataset>
> ```

### D.2    Details of Other Experiments

In Figure 2, we discussed the success probability of BoN, which requires an estimate of BoN's output distribution. We use the same technique as in MoB to estimate this output distribution. To minimize the error of this approximation, we specifically generate 1,400 outputs for the math problems in MMLU-Pro with Qwen2.5-3B . Then, we use $\hat{\pi}_{N,1400}$, as defined in Section 4.2 as an estimate for $\pi_N$. Same technique is used in Figure 3 where the mode of $\hat{\pi}_{N,1400}$ is chosen as the output of oracle MoB, and Figure 4 to where the distribution estimation error is calculated with respect to $\hat{\pi}_{m,1400}$ instead of the true $\pi_m$.

In Figure 5, we consider seven fixed schedules for $m$, specifically $m = \lfloor N^\alpha \rfloor$ for $\alpha = 0.2, 0.3, 0.4, 0.5, 0.6, 0.7, 0.8$. At any budget $N$, we compared the accuracy of MoB with adaptive $m$ against the highest accuracy among the seven instantiations of fixed schedule MoB.

In Figure 6, for each question, we measure base model's success probability and reward model's accuracy using 512 outputs. We ignore questions with all-correct or all-incorrect outputs, since the

reward accuracy is not defined for them, as well as questions with reward accuracy bellow $0.25$ due to all algorithms having zero success probability on them. Also, $m$ is calculated adaptively for each question.

# E Additional Experimental Results

In this section, we provide additional experimental results for all 30 setups.

## E.1 Adaptive Subset Size Selection

In Section 4, we compared MoB with adaptive choice of $m$ and $m = \sqrt{N}$ with the optimal choice of $m$. We provide this comparison in MATH500 (Figure 9), MMLU-Pro-Math (Figure 10), MMLU-Pro-Chem (Figure 11), GSM8K (Figure 12), and CommonsenseQA (Figure 13). In Table 3, we compare the performance of MoB with adaptive $q$ for various values of $q$ on all benchmarks Llama3.1-8B base model and ArmoRM reward model. As also observed in the literature, we observe that the choice of $q$ is not a sensitive one.

Table 3: Performance of MoB with adaptive $m$ across different choices of $q$ for Llama3.1-8B base model and ArmoRM reward model.

| $q$ | 0.40 | 0.50 | 0.60 | 0.70 | 0.80 | 0.90 |
|---|---|---|---|---|---|---|
| MATH500 | 61.85% | 63.00% | 63.15% | 62.20% | 62.45% | 60.70% |
| MMLU-Pro-Math | 66.60% | 66.85% | 66.75% | 66.60% | 67.10% | 66.35% |
| MMLU-Pro-Chem | 56.75% | 57.40% | 57.85% | 57.15% | 56.95% | 56.35% |
| GSM8k | 91.60% | 91.55% | 91.55% | 91.85% | 91.80% | 91.85% |
| CSQA | 77.45% | 77.40% | 77.30% | 77.25% | 77.35% | 77.30% |

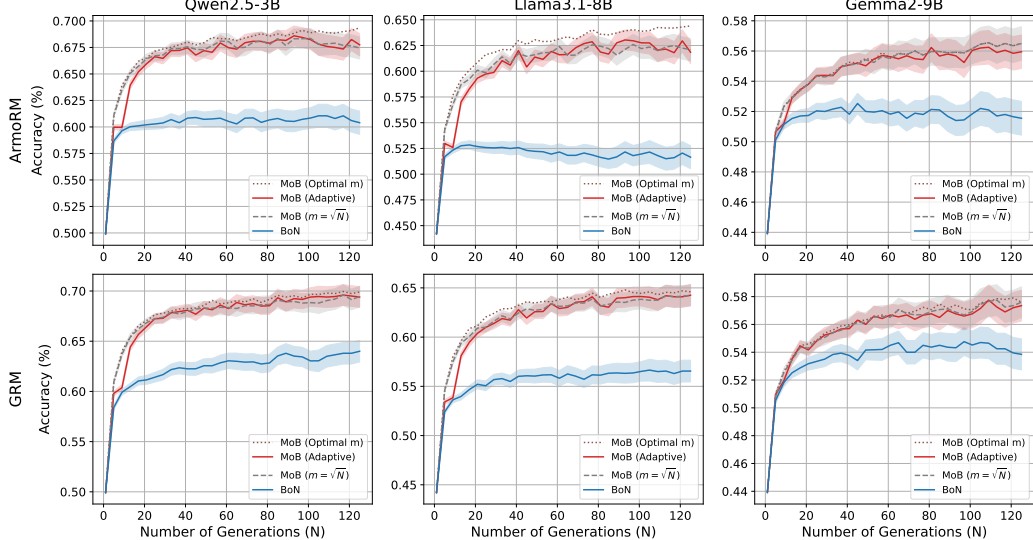

Figure 9: Comparison of MoB with adaptive $m$ and $m = \sqrt{N}$ against MoB with optimal $m$ on the MATH500 dataset with ArmoRM *(Up)* and GRM *(Down)* reward models, and Qwen2.5-3B *(Left)*, Llama3.1-8B *(Middle)*, and Gemma2-9B *(Right)* base models. Shaded areas show standard error.

## E.2 Evaluation Experiments

We compare MoB with adaptive $m$ and $m = \sqrt{N}$ with baselines in MATH500 (Figure 14, Table 4), MMLU-Pro-Math (Figure 15, Table 5), MMLU-Pro-Chem (Figure 16, Table 6), GSM8K (Figure 17, Table 7), and CommonsenseQA (Figure 18, Table 8).

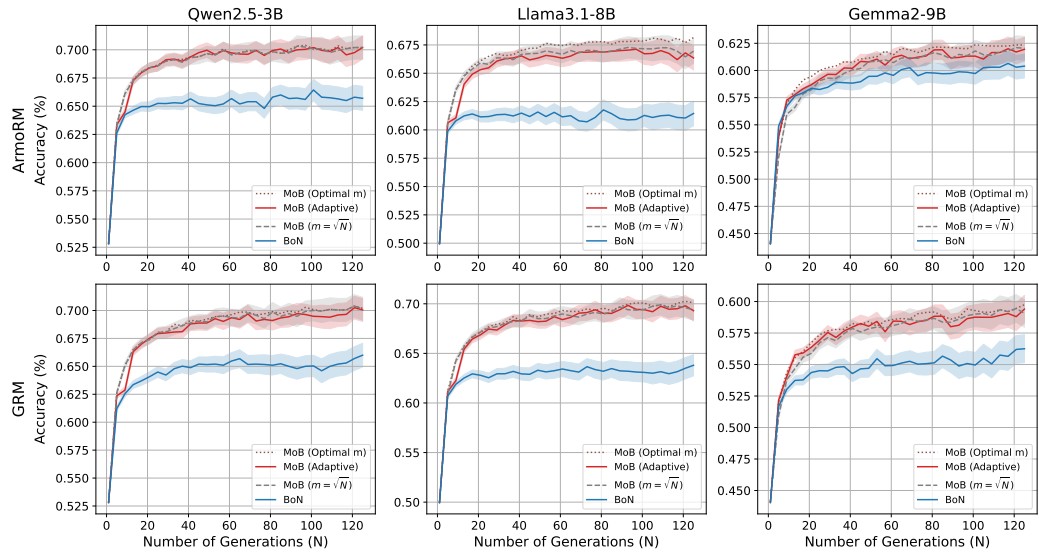

Figure 10: Comparison of MoB with adaptive $m$ and $m = \sqrt{N}$ against MoB with optimal $m$ on the MMLU-Pro-Math dataset with ArmoRM *(Up)* and GRM *(Down)* reward models, and Qwen2.5-3B *(Left)*, Llama3.1-8B *(Middle)*, and Gemma2-9B *(Right)* base models. Shaded areas show standard error.

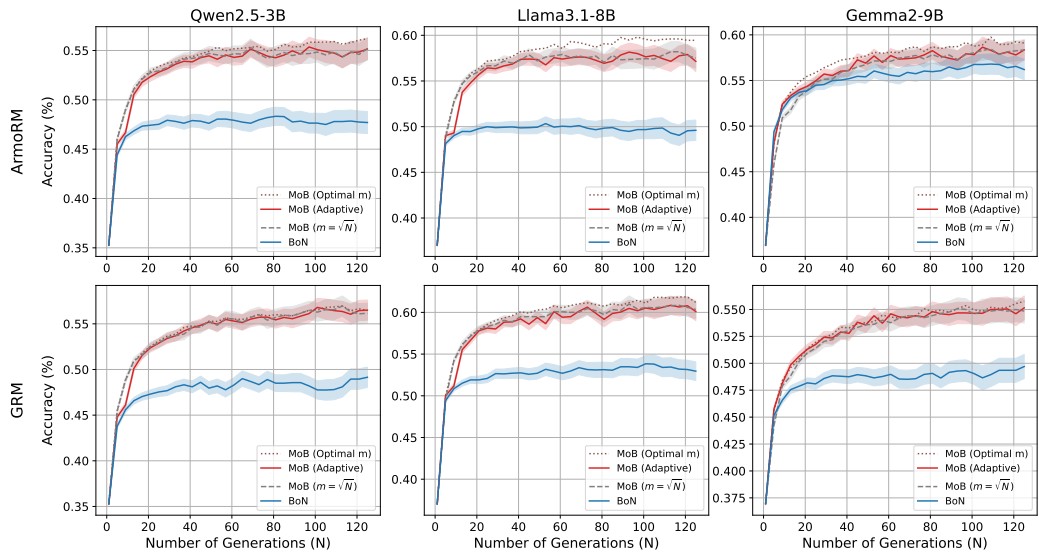

Figure 11: Comparison of MoB with adaptive $m$ and $m = \sqrt{N}$ against MoB with optimal $m$ on the MMLU-Pro-Chem dataset with ArmoRM *(Up)* and GRM *(Down)* reward models, and Qwen2.5-3B *(Left)*, Llama3.1-8B *(Middle)*, and Gemma2-9B *(Right)* base models. Shaded areas show standard error.

### E.3 Results on Skywork-v2 Reward Model

Additionally, we report the results for `Skywork/Skywork-Reward-V2-Llama-3.1-8B` (Liu et al., 2025), a more recent reward model in Tables 9-13.

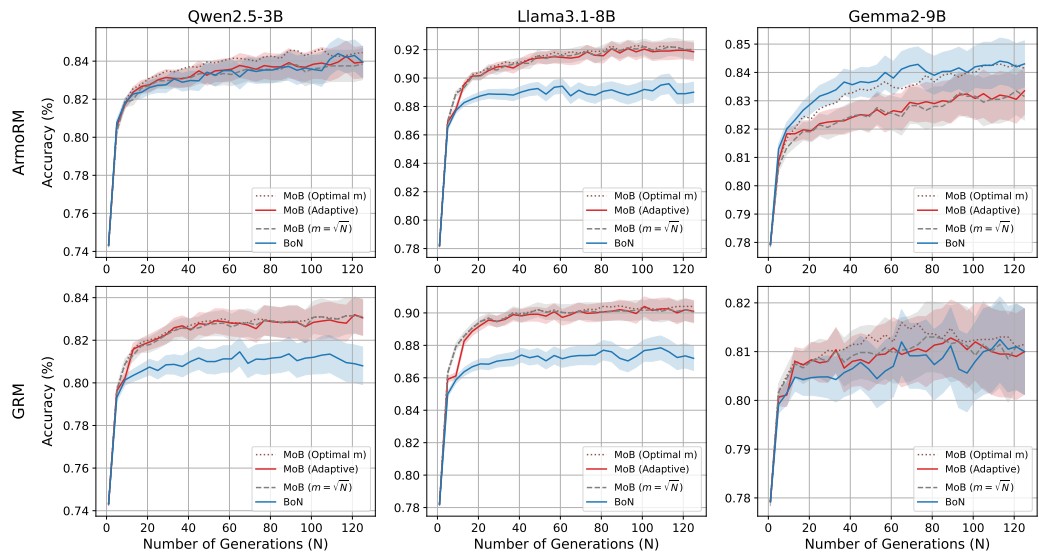

Figure 12: Comparison of MoB with adaptive $m$ and $m = \sqrt{N}$ against MoB with optimal $m$ on the GSM8K dataset with ArmoRM *(Up)* and GRM *(Down)* reward models, and Qwen2.5-3B *(Left)*, Llama3.1-8B *(Middle)*, and Gemma2-9B *(Right)* base models. Shaded areas show standard error.

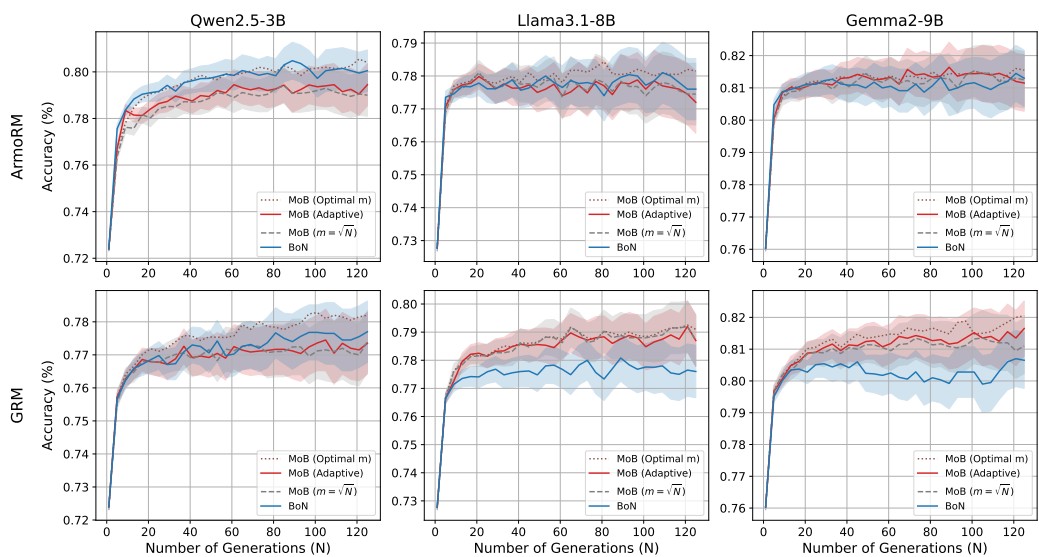

Figure 13: Comparison of MoB with adaptive $m$ and $m = \sqrt{N}$ against MoB with optimal $m$ on the CommonsenseQA dataset with ArmoRM *(Up)* and GRM *(Down)* reward models, and Qwen2.5-3B *(Middle)*, and Gemma2-9B *(Right)* base models. Shaded areas show standard error.

Table 4: Results on MATH500 across all base and reward models ($N = 128$).

| | **ArmoRM** | | | **GRM** | | |
|---|---|---|---|---|---|---|
| | Llama3.1-8B | Gemma2-9B | Qwen2.5-3B | Llama3.1-8B | Gemma2-9B | Qwen2.5-3B |
| BoN | $51.55_{\pm1.12}$ | $52.20_{\pm1.12}$ | $60.60_{\pm1.09}$ | $56.65_{\pm1.11}$ | $54.95_{\pm1.11}$ | $63.95_{\pm1.07}$ |
| SC | $60.65_{\pm1.09}$ | $52.90_{\pm1.12}$ | $66.40_{\pm1.06}$ | $60.65_{\pm1.09}$ | $52.90_{\pm1.12}$ | $66.40_{\pm1.06}$ |
| WBoN | $\mathbf{62.90_{\pm1.08}}$ | $53.85_{\pm1.11}$ | $67.10_{\pm1.05}$ | $\mathbf{63.55_{\pm1.08}}$ | $56.15_{\pm1.11}$ | $67.45_{\pm1.05}$ |
| MoB-Adaptive (Ours) | $\mathbf{62.90_{\pm1.08}}$ | $56.15_{\pm1.11}$ | $\mathbf{68.50_{\pm1.04}}$ | $\mathbf{64.30_{\pm1.07}}$ | $57.45_{\pm1.11}$ | $\mathbf{69.95_{\pm1.03}}$ |
| MoB-Poly (Ours) | $62.40_{\pm1.08}$ | $\mathbf{57.05_{\pm1.11}}$ | $67.85_{\pm1.04}$ | $64.00_{\pm1.07}$ | $\mathbf{58.10_{\pm1.10}}$ | $69.45_{\pm1.03}$ |
| ↑MoB over BoN | $\underline{11.35_{\pm0.86}}$ | $\underline{3.95_{\pm0.68}}$ | $\underline{7.90_{\pm0.78}}$ | $\underline{7.65_{\pm0.80}}$ | $\underline{2.50_{\pm0.64}}$ | $\underline{6.00_{\pm0.78}}$ |

Table 5: Results on MMLU-Pro-Math across all base and reward models ($N = 128$).

| | ArmoRM | | | GRM | | |
|---|---|---|---|---|---|---|
| | Llama3.1-8B | Gemma2-9B | Qwen2.5-3B | Llama3.1-8B | Gemma2-9B | Qwen2.5-3B |
| BoN | $61.40_{\pm1.09}$ | $60.45_{\pm1.09}$ | $65.95_{\pm1.06}$ | $64.10_{\pm1.07}$ | $56.15_{\pm1.11}$ | $66.10_{\pm1.06}$ |
| SC | $62.95_{\pm1.08}$ | $49.95_{\pm1.12}$ | $65.60_{\pm1.06}$ | $62.95_{\pm1.08}$ | $49.95_{\pm1.12}$ | $65.60_{\pm1.06}$ |
| WBoN | $\mathbf{66.45_{\pm1.06}}$ | $52.25_{\pm1.12}$ | $66.70_{\pm1.05}$ | $60.05_{\pm1.10}$ | $56.45_{\pm1.11}$ | $64.35_{\pm1.07}$ |
| MoB-Adaptive (Ours) | $\mathbf{66.70_{\pm1.05}}$ | $\mathbf{61.55_{\pm1.09}}$ | $\mathbf{69.80_{\pm1.03}}$ | $\mathbf{69.05_{\pm1.03}}$ | $59.35_{\pm1.10}$ | $69.30_{\pm1.03}$ |
| MoB-Poly (Ours) | $\mathbf{67.20_{\pm1.05}}$ | $\mathbf{62.05_{\pm1.09}}$ | $\mathbf{70.05_{\pm1.02}}$ | $\mathbf{69.30_{\pm1.03}}$ | $59.45_{\pm1.10}$ | $70.15_{\pm1.02}$ |
| ↑MoB over BoN | $\underline{5.30_{\pm0.81}}$ | $\underline{1.10_{\pm0.71}}$ | $\underline{3.85_{\pm0.80}}$ | $\underline{4.95_{\pm0.82}}$ | $\underline{3.20_{\pm0.83}}$ | $\underline{3.20_{\pm0.79}}$ |

Table 6: Results on MMLU-Pro-Chem across all base and reward models ($N = 128$).

| | ArmoRM | | | GRM | | |
|---|---|---|---|---|---|---|
| | Llama3.1-8B | Gemma2-9B | Qwen2.5-3B | Llama3.1-8B | Gemma2-9B | Qwen2.5-3B |
| BoN | $49.70_{\pm1.12}$ | $56.60_{\pm1.11}$ | $48.05_{\pm1.12}$ | $53.05_{\pm1.12}$ | $49.25_{\pm1.12}$ | $49.00_{\pm1.12}$ |
| SC | $50.25_{\pm1.12}$ | $43.40_{\pm1.11}$ | $52.50_{\pm1.12}$ | $50.25_{\pm1.12}$ | $43.40_{\pm1.11}$ | $52.50_{\pm1.12}$ |
| WBoN | $\mathbf{57.65_{\pm1.10}}$ | $45.45_{\pm1.11}$ | $53.30_{\pm1.12}$ | $49.75_{\pm1.12}$ | $\mathbf{57.25_{\pm1.11}}$ | $53.10_{\pm1.12}$ |
| MoB-Adaptive (Ours) | $\mathbf{57.40_{\pm1.11}}$ | $58.05_{\pm1.10}$ | $\mathbf{54.75_{\pm1.11}}$ | $\mathbf{60.75_{\pm1.09}}$ | $54.60_{\pm1.11}$ | $\mathbf{56.45_{\pm1.11}}$ |
| MoB-Poly (Ours) | $\mathbf{57.80_{\pm1.10}}$ | $58.80_{\pm1.10}$ | $\mathbf{54.90_{\pm1.11}}$ | $60.00_{\pm1.10}$ | $55.00_{\pm1.11}$ | $\mathbf{56.30_{\pm1.11}}$ |
| ↑MoB over BoN | $\underline{7.70_{\pm0.92}}$ | $\underline{1.45_{\pm0.80}}$ | $\underline{6.70_{\pm0.92}}$ | $\underline{7.70_{\pm0.93}}$ | $\underline{5.35_{\pm0.92}}$ | $\underline{7.45_{\pm0.94}}$ |

Table 7: Results on GSM8K across all base and reward models ($N = 128$).

| | ArmoRM | | | GRM | | |
|---|---|---|---|---|---|---|
| | Llama3.1-8B | Gemma2-9B | Qwen2.5-3B | Llama3.1-8B | Gemma2-9B | Qwen2.5-3B |
| BoN | $89.00_{\pm0.70}$ | $\mathbf{84.20_{\pm0.82}}$ | $\mathbf{83.85_{\pm0.82}}$ | $87.15_{\pm0.75}$ | $\mathbf{81.20_{\pm0.87}}$ | $80.95_{\pm0.88}$ |
| SC | $88.15_{\pm0.72}$ | $80.55_{\pm0.89}$ | $80.40_{\pm0.89}$ | $88.15_{\pm0.72}$ | $\mathbf{80.55_{\pm0.89}}$ | $80.40_{\pm0.89}$ |
| WBoN | $88.70_{\pm0.71}$ | $80.75_{\pm0.88}$ | $81.10_{\pm0.88}$ | $77.75_{\pm0.93}$ | $79.45_{\pm0.90}$ | $81.25_{\pm0.87}$ |
| MoB-Adaptive (Ours) | $\mathbf{91.75_{\pm0.62}}$ | $83.30_{\pm0.83}$ | $\mathbf{83.85_{\pm0.82}}$ | $\mathbf{90.50_{\pm0.66}}$ | $81.15_{\pm0.87}$ | $82.85_{\pm0.84}$ |
| MoB-Poly (Ours) | $\mathbf{91.80_{\pm0.61}}$ | $83.15_{\pm0.84}$ | $\mathbf{83.80_{\pm0.82}}$ | $90.05_{\pm0.67}$ | $80.85_{\pm0.88}$ | $83.10_{\pm0.84}$ |
| ↑MoB over BoN | $\underline{2.75_{\pm0.56}}$ | $\underline{-0.90_{\pm0.52}}$ | $\underline{0.00_{\pm0.50}}$ | $\underline{3.35_{\pm0.56}}$ | $\underline{-0.05_{\pm0.47}}$ | $\underline{1.90_{\pm0.51}}$ |

Table 8: Results on CSQA across all base and reward models ($N = 128$).

| | ArmoRM | | | GRM | | |
|---|---|---|---|---|---|---|
| | Llama3.1-8B | Gemma2-9B | Qwen2.5-3B | Llama3.1-8B | Gemma2-9B | Qwen2.5-3B |
| BoN | $\mathbf{77.80_{\pm0.93}}$ | $\mathbf{81.20_{\pm0.87}}$ | $\mathbf{80.15_{\pm0.89}}$ | $78.05_{\pm0.93}$ | $80.55_{\pm0.89}$ | $\mathbf{77.70_{\pm0.93}}$ |
| SC | $75.75_{\pm0.96}$ | $79.25_{\pm0.91}$ | $76.20_{\pm0.95}$ | $75.75_{\pm0.96}$ | $79.25_{\pm0.91}$ | $76.20_{\pm0.95}$ |
| WBoN | $76.75_{\pm0.94}$ | $80.05_{\pm0.89}$ | $76.60_{\pm0.95}$ | $36.35_{\pm1.08}$ | $49.80_{\pm1.12}$ | $54.90_{\pm1.11}$ |
| MoB-Adaptive (Ours) | $77.40_{\pm0.94}$ | $\mathbf{81.20_{\pm0.87}}$ | $79.40_{\pm0.90}$ | $78.45_{\pm0.92}$ | $\mathbf{81.45_{\pm0.87}}$ | $77.40_{\pm0.94}$ |
| MoB-Poly (Ours) | $77.30_{\pm0.94}$ | $\mathbf{81.45_{\pm0.87}}$ | $79.15_{\pm0.91}$ | $\mathbf{78.65_{\pm0.92}}$ | $81.15_{\pm0.87}$ | $77.45_{\pm0.93}$ |
| ↑MoB over BoN | $\underline{-0.40_{\pm0.47}}$ | $\underline{0.00_{\pm0.43}}$ | $\underline{-0.75_{\pm0.48}}$ | $\underline{0.40_{\pm0.54}}$ | $\underline{0.90_{\pm0.48}}$ | $\underline{-0.30_{\pm0.52}}$ |

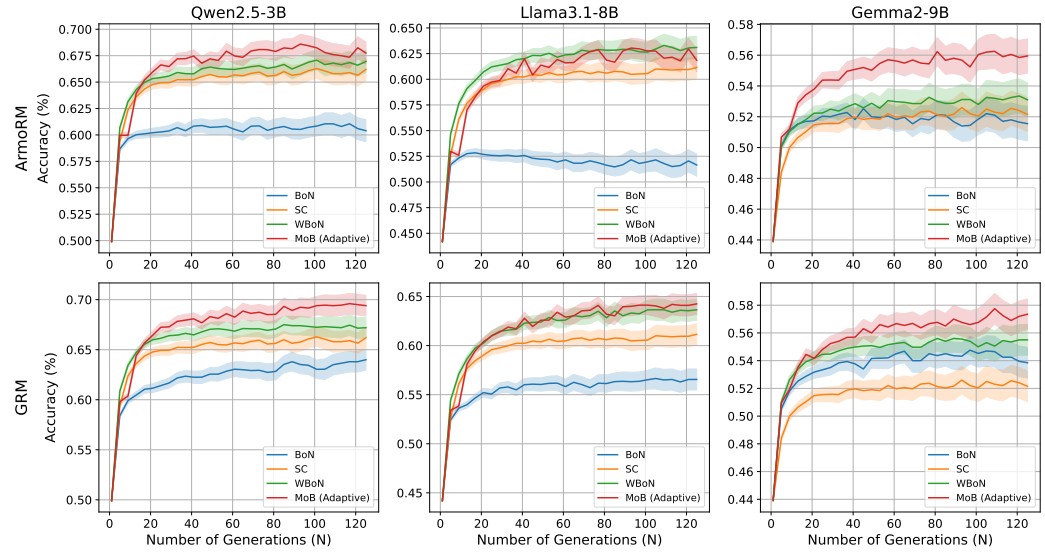

Figure 14: Comparison of MoB with the baselines on the MATH500 dataset with ArmoRM *(Up)* and GRM *(Down)* reward models, and Qwen2.5-3B *(Left)*, Llama3.1-8B *(Middle)*, and Gemma2-9B *(Right)* base models. Shaded areas show standard error.

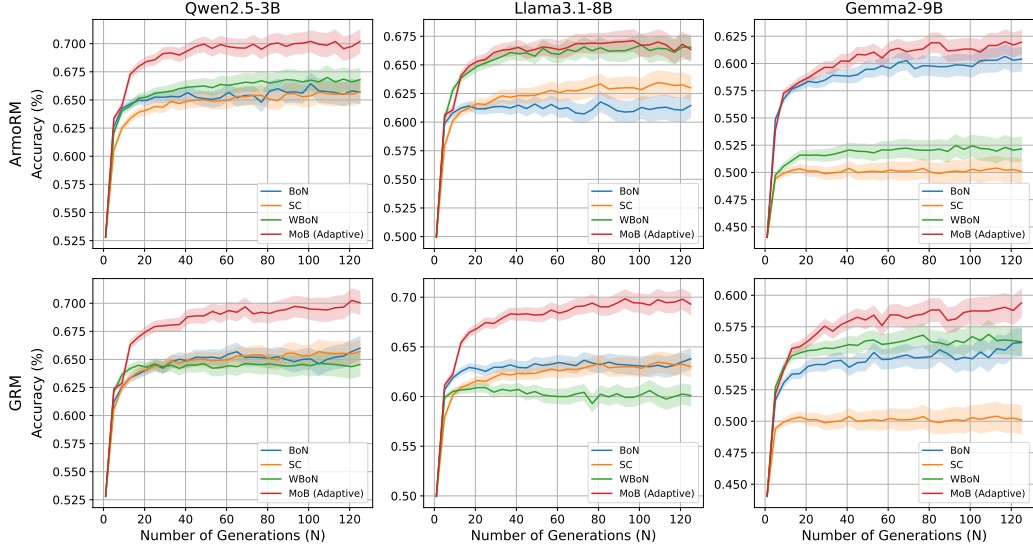

Figure 15: Comparison of MoB with the baselines on the MMLU-Pro-Math dataset with ArmoRM *(Up)* and GRM *(Down)* reward models, and Qwen2.5-3B *(Left)*, Llama3.1-8B *(Middle)*, and Gemma2-9B *(Right)* base models. Shaded areas show standard error.

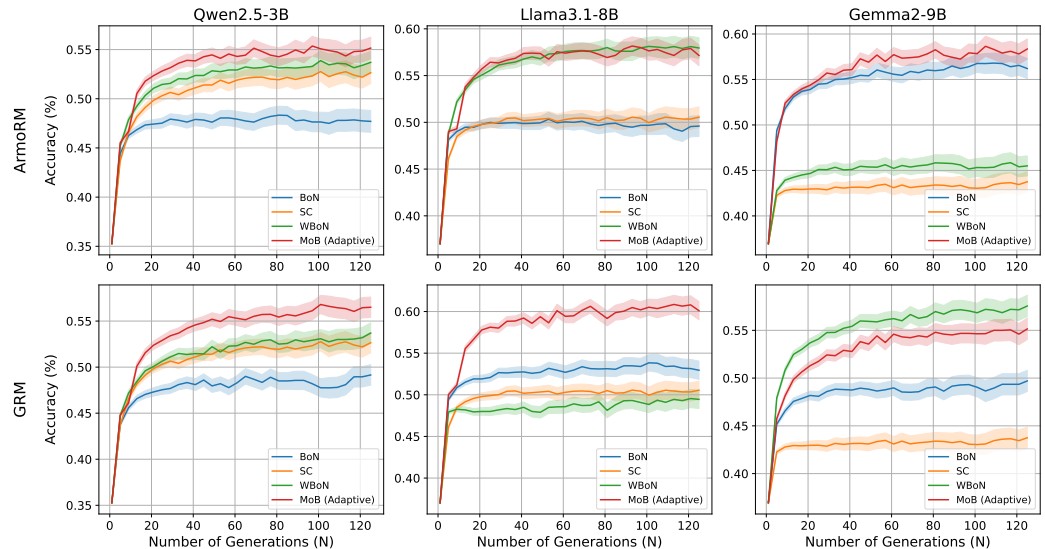

Figure 16: Comparison of MoB with the baselines on the MMLU-Pro-Chem dataset with ArmoRM *(Up)* and GRM *(Down)* reward models, and Qwen2.5-3B *(Left)*, Llama3.1-8B *(Middle)*, and Gemma2-9B *(Right)* base models. Shaded areas show standard error.

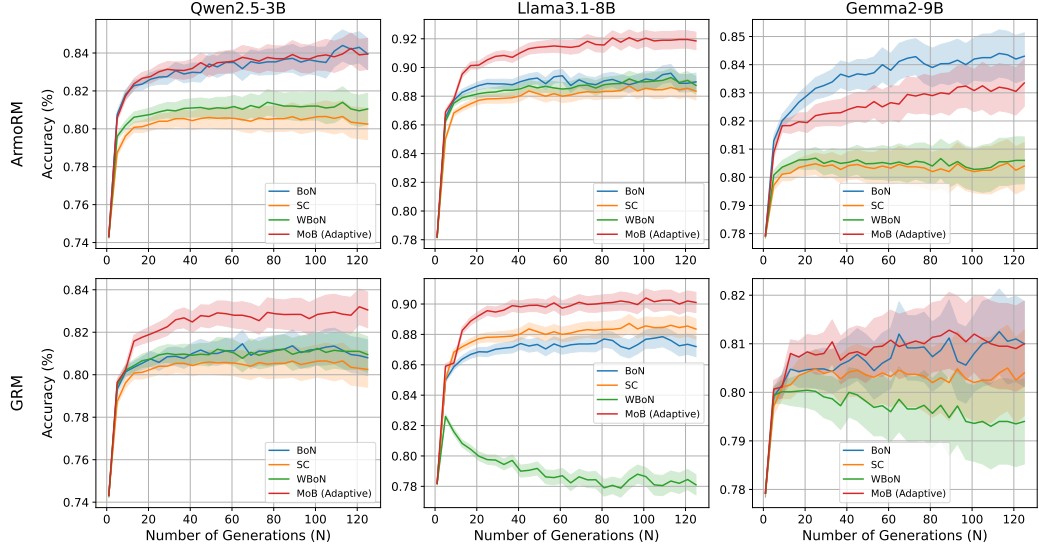

Figure 17: Comparison of MoB with the baselines on the GSM8K dataset with ArmoRM *(Up)* and GRM *(Down)* reward models, and Qwen2.5-3B *(Left)*, Llama3.1-8B *(Middle)*, and Gemma2-9B *(Right)* base models. Shaded areas show standard error.

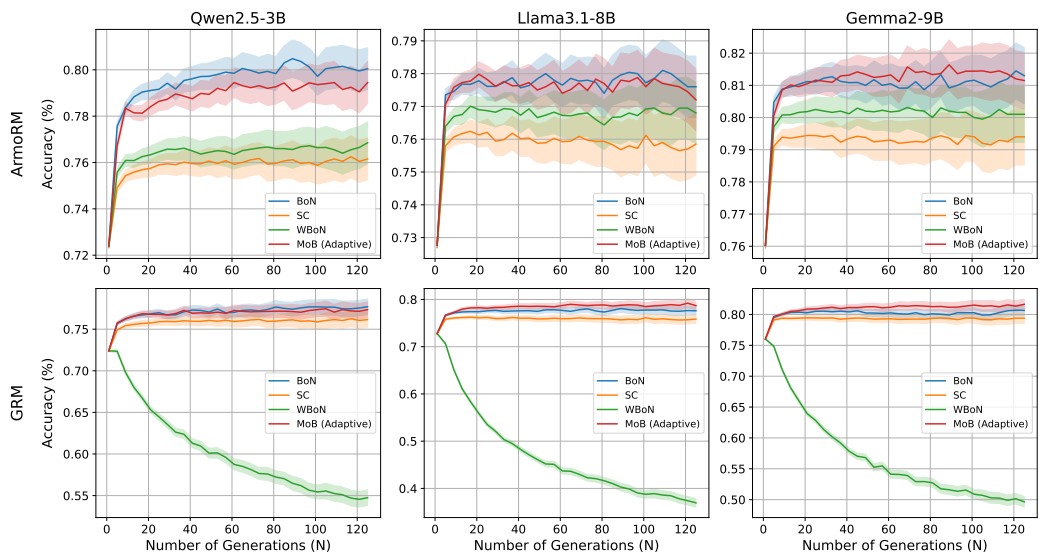

Figure 18: Comparison of MoB with the baselines on the CommonsenseQA dataset with ArmoRM *(Up)* and GRM *(Down)* reward models, and Qwen2.5-3B *(Left)*, Llama3.1-8B *(Middle)*, and Gemma2-9B *(Right)* base models. Shaded areas show standard error.

Table 9: Results on MATH500 with Skywork reward model across all base models ($N = 128$).

|  | **Skywork** | | |
|  | Llama3.1-8B | Gemma2-9B | Qwen2.5-3B |
| --- | --- | --- | --- |
| BoN | 54.50±0.79 | 55.85±0.79 | 63.35±0.76 |
| SC | 60.65±1.09 | 52.90±1.12 | 66.40±1.06 |
| WBoN | **65.05±1.07** | 57.95±1.10 | 68.80±1.04 |
| MoB-Adaptive (Ours) | 63.95±1.07 | **59.45±1.10** | 69.70±1.03 |
| MoB-Poly (Ours) | 63.65±1.08 | **59.15±1.10** | **70.35±1.02** |
| ↑MoB over BoN | 9.45±0.81 | 3.60±0.72 | 6.35±0.78 |

Table 10: Results on MMLU-Pro-Math with Skywork reward model across all base models ($N = 128$).

|  | **Skywork** | | |
|  | Llama3.1-8B | Gemma2-9B | Qwen2.5-3B |
| --- | --- | --- | --- |
| BoN | 60.10±0.77 | 54.20±0.79 | 67.80±0.74 |
| SC | 62.95±1.08 | 49.95±1.12 | 65.60±1.06 |
| WBoN | **69.45±1.03** | **60.00±1.10** | 69.80±1.03 |
| MoB-Adaptive (Ours) | 66.00±1.06 | **59.10±1.10** | **72.55±1.00** |
| MoB-Poly (Ours) | 66.70±1.05 | **59.10±1.10** | **72.85±0.99** |
| ↑MoB over BoN | 5.90±0.75 | 4.90±0.88 | 4.75±0.75 |

Table 11: Results on MMLU-Pro-Chem with Skywork reward model across all base models ($N = 128$).

| | Skywork | | |
| --- | --- | --- | --- |
| | Llama3.1-8B | Gemma2-9B | Qwen2.5-3B |
| BoN | 57.23±0.82 | 53.20±0.79 | 57.70±0.78 |
| SC | 50.60±1.17 | 43.40±1.11 | 52.50±1.12 |
| WBoN | **62.83±1.13** | **58.60±1.10** | 58.65±1.10 |
| MoB-Adaptive (Ours) | 60.87±1.14 | **57.75±1.10** | 61.50±1.09 |
| MoB-Poly (Ours) | 60.92±1.14 | **57.40±1.11** | 61.55±1.09 |
| ↑MoB over BoN | 3.64±0.79 | 4.55±0.98 | 3.80±0.82 |

Table 12: Results on GSM8K with Skywork reward model across all base models ($N = 128$).

| | Skywork | | |
| --- | --- | --- | --- |
| | Llama3.1-8B | Gemma2-9B | Qwen2.5-3B |
| BoN | 85.15±0.56 | **80.98±0.62** | 82.37±1.05 |
| SC | 88.15±0.72 | 80.53±0.89 | 80.85±1.53 |
| WBoN | 88.25±0.72 | 80.13±0.89 | 82.07±1.50 |
| MoB-Adaptive (Ours) | **89.55±0.68** | **81.23±0.87** | **84.04±1.43** |
| MoB-Poly (Ours) | **89.85±0.68** | **81.23±0.87** | **83.89±1.43** |
| ↑MoB over BoN | 4.40±0.61 | 0.25±0.46 | 1.67±0.82 |

Table 13: Results on CSQA with Skywork reward model across all base models ($N = 128$).

| | Skywork | | |
| --- | --- | --- | --- |
| | Llama3.1-8B | Gemma2-9B | Qwen2.5-3B |
| BoN | 77.00±0.67 | 80.15±0.63 | **78.85±0.65** |
| SC | 75.75±0.96 | 79.25±0.91 | 76.20±0.95 |
| WBoN | 76.45±0.95 | **80.20±0.89** | 76.15±0.95 |
| MoB-Adaptive (Ours) | **77.80±0.93** | **81.00±0.88** | 77.00±0.94 |
| MoB-Poly (Ours) | **78.00±0.93** | **81.00±0.88** | 77.25±0.94 |
| ↑MoB over BoN | 0.80±0.43 | 0.85±0.41 | -1.85±0.49 |

