# OpenReview forum: "Majority of the Bests: Improving Best-of-N via Bootstrapping"
_NeurIPS.cc/2025/Conference — NeurIPS 2025 poster_

### Official Review · Reviewer_DAFe · 2025-06-10

**Clarity:** 2
**Significance:** 3
**Originality:** 2
**Rating:** 4
**Confidence:** 4

**Summary:**

A popular trend in inference-time scaling of LLMs involves sampling many responses, and using some strategy to aggregate their answers to produce a single final answer.  This work is motivated by the shortcomings of common approaches like Best-of-N (BoN) and self-consistency (SC), proposing to improve on BoN by applying a bootstrap approach to debiasing the choice of the noisy reward model.  Theoretical and empirical evidence is provided to support the approach, with experiments showcasing applications in Math, Chemistry and QA.

**Questions:**

Please see weaknesses section.

**Ethical Concerns:**

["NO or VERY MINOR ethics concerns only"]

**Final Justification:**

The authors did a great job in their rebuttal of addressing the various "why" questions that I had about the experiments.  As such, I revised my score upward to a weak accept.

**Limitations:**

The authors address limitations in one sentence in the conclusion.  More discussion (and demonstration) of the algorithm’s limitations would be welcomed.

**Quality:**

2

**Strengths And Weaknesses:**

## Strengths

This paper offers a solution to a high-priority problem for the LLM community: broadly, what is the best way to scale inference-time compute, and in particular, how should we combine solutions from many samples.  As opposed to the many heuristic approaches that have been taken to this problem, the bootstrapping method proposed here is underpinned by rigorous theory. Also, the compute cost compares favorably with other methods.

In terms of the paper presentation, the method description (and accompanying empirical demonstrations) offer solid support for the proposed algorithm.

## Weaknesses

While I appreciate the simplicity and motivation of the MoB algorithm, I find the experimental evaluation to have significant weaknesses that prevent the reader from gaining a thorough impression of when and why this algorithm might be helpful.

First, it is not clear why certain experiment details are chosen.  For example, why is evaluation done on this selection of 3 math, 1 chemistry, and 1 QA tasks?  Are these domains where you expect the method to be particularly useful, or that are helpful to reveal the strengths and weaknesses of the algorithm?  Also, does it make sense to use these two reward models for all of these tasks?  One could imagine that different reward models might be chosen for different tasks, e.g., math or QA.  Also, these do not seem to be the strongest available reward models of their size (according to rewardbench).  Finally, why do some experiments use few-shot prompting, and some use zero-shot CoT?

Next, related to the last question, it seems that baseline implementations may not match the approach suggested in the original work.  For Wang 2022, I believe that SC is always meant to be combined with chain of thought (as opposed to few-shot prompting).  Similarly, for WBoN with Li 2022 as a reference, I believe the original method involves some diverse prompting and (crucially) step-level labeling, which do not seem to be applied here.

Finally, and potentially my biggest concern with the experiments, is that the presentation of results seems aimed at boosting the perceived performance of MoB.  For example, I am quite confused by how items are bolded or not in tables.  Across the 5 tables, I found 9 examples of unbolded results that have overlapping CIs with the best result.  Accordingly, I am further confused by the claim in the conclusion that MoB outperforms all baselines in 25 of 30 settings.  This does not reflect the results in the tables.

Overall, I feel that the experiment section can benefit from more focus on “when” and “why?” questions about the algorithm’s performance, as opposed to focusing purely on outperforming baselines.  Given that the algorithm is not technically deep, the weaknesses of the experimental results outweigh the strengths of the paper in my opinion.

Also, ideally code would be included.

---

> ### Author Rebuttal · Authors · 2025-07-31
>
> We thank the reviewer for their thorough review with close attention to details of our work. We would like to present further analysis on when MoB work and clarify some of the concerns.
>
> # When does MoB succeed/fail?
> The change from BoN to MoB can be split into two **steps**:
>
> 1. **Change from Best-of-$N$ to Best-of-$m$**: To be able to approximate the distribution, MoB is forced to work with the output distribution of Best-of-$m$ ($\pi_m$) for some $m < N$. This is usually inferior to $\pi_N$ used by BoN, which means this step is a downgrade in accuracy.
>
> 2. **Change from Best-of-$m$ to MoB**: MoB estimate and picks the mode of $\pi_m$, but Best-of-$m$ samples from it. If the mode is correct, MoB will be correct with high probability and improve the success probability over Best-of-$m$. On the other hand if the mode is not correct, the small probability to solve the problem by chance is eliminated and MoB will be worse.
>
> Together, the impacts of these two steps on the performance govern whether MoB will outperform BoN or not. The impact of each of these steps varies for each question and depends on the base model's generation and reward model's reward distributions in that question. These two distributions, especially the reward, can be complex. We suggest the following two scalar metrics to allow further intuitive analysis.
>
> * BaseSProb = Chance of a generation being correct.
> * RewardAcc = Fraction of correct-incorrect output pairs correctly judged by the reward model.
>
> These two metrics can be estimated for each question and evaluate how good the base and reward models are for that question. We now study how these metrics can predict the impact of the modification steps from BoN to MoB discussed above. Then, we study when MoB can improve over BoN.
>
> ## Impact of the change from Best-of-$N$ to Best-of-$m$ (Step 1)
> Consider the setup with MMLU-Pro-Math, gemma2-9b-it, and ArmoRM. To match our experiments section, we set $N = 128$. The following table shows the accuracy of Best-of-$N$ minus the accuracy of Best-of-$m$ for subsets of the benchmark with different reward and base model accuracies. Note that the value of $m$ varies among questions.
>
> |**Reward Acc \ Base S Prob**|$0.00-0.25$|$0.25-0.50$|$0.50-0.75$|$0.75-1.00$|
> |:-:|-:|-:|-:|-:|
> |  $0.75-1.00$ |  20 |  13 | 8 | 0 |
> |  $0.50-0.75$ | 5 | 5 |  -0 | 0 |
> |  $0.25-0.50$ |  -2 | -22 | -14 | -32 |
>
> The smaller the advantage of Best-of-$N$ over Best-of-$m$ shown in the table is, the less MoB suffers from its use of $\pi_m$ instead of $\pi_N$. The results of the table match the intuition, increase in budge from $m$ to $N$ is more beneficial in questions with good rewards (top row). Especially when the chance of generating the correct answer is small, we benefit a lot by having a larger number of generations $N$ compared to $m$ (top-left cell). **MoB is negatively affected if the BoN performance is not saturated for a question and more budget significantly improves BoN performance.**
>
> ## Impact of the change from Best-of-$m$ to MoB (Step 2)
> We can categorize questions into three categories:
>
> * Type A: Questions where mode of $\pi_m$ is correct, and $0.1 \le \pi_m(Z^*) \le 0.9$. Picking the mode is better than sampling.
> * Type B: Questions where mode of $\pi_m$ is incorrect, and $0.1 \le \pi_m(Z^*) \le 0.9$. Sampling is better than picking the mode.
> * Type C: Questions where $ \pi_m(Z^*) > 0.9 $ or $ \pi_m(Z^*) < 0.1$ . Sampling and picking the mode are similar.
>
> The impact of this step can roughly be measured by the number of questions of type A and B. Higher number of type A questions have a positive effect on the performance of MoB. Table 2 shows the frequency of these types in each combination of generation and reward accuracy.
>
> |  **Reward Acc \ BaseSProb**  |  $0.00-0.25$  |  $0.25-0.50$  | $0.50-0.75$  | $0.75-1.00$  |
> |:-:|:-:|:-:|:-:|:-:|
> | $0.75-1.00$  | $ \textcolor{blue}{A: 33}, \textcolor{red}{B: 33} $ | $ \textcolor{blue}{\textbf{A: 13}}, \textcolor{red}{B: 0} $ | $ \textcolor{blue}{\textbf{A: 5}}, \textcolor{red}{B: 2} $ | $ \textcolor{blue}{\textbf{A: 1}}, \textcolor{red}{B: 0} $ |
> | $0.50-0.75$  | $ \textcolor{blue}{A: 4}, \textcolor{red}{\textbf{B: 13}} $ | $ \textcolor{blue}{\textbf{A: 14}}, \textcolor{red}{B: 7} $ | $ \textcolor{blue}{\textbf{A: 8}}, \textcolor{red}{B: 0} $ | $ \textcolor{blue}{\textbf{A: 4}}, \textcolor{red}{B: 0} $ |
> | $0.25-0.50$  | - | $ \textcolor{blue}{A: 1}, \textcolor{red}{\textbf{B: 2}} $  | $ \textcolor{blue}{A: 1}, \textcolor{red}{B: 1} $  | $ \textcolor{blue}{A: 0}, \textcolor{red}{\textbf{B: 1}} $ |
>
> **In questions with poor generation or rewards, the mode is incorrect more often. In comparison to BoN, MoB has a smaller probability to solve these questions by chance.**
>
> ## Comparing MoB and BoN under various generation and reward accuracies
> The relative performance of MoB to BoN depends on the combination of the impacts of the two steps. The following table gives the final accuracies.
>
> |  **Reward Acc \ Base S Prob** |$0.00-0.25$|$0.25-0.50$ |  $0.50-0.75$ |  $0.75-1.00$ |
> |:-:|:-:|:-:|:-:|:-:|
> |  $0.75-1.00$ | $ \textcolor{blue}{MoB: 38}, \textcolor{red}{\textbf{BoN: 45}} $ | $ \textcolor{blue}{\textbf{MoB: 97}}, \textcolor{red}{BoN: 83} $ | $ \textcolor{blue}{\textbf{MoB: 99}}, \textcolor{red}{BoN: 95} $ |  $ \textcolor{blue}{MoB: 100}, \textcolor{red}{BoN: 100} $ |
> |  $0.50-0.75$ | $ \textcolor{blue}{MoB: 8}, \textcolor{red}{\textbf{BoN: 18}} $  | $ \textcolor{blue}{\textbf{MoB: 68}}, \textcolor{red}{BoN: 53} $ | $ \textcolor{blue}{\textbf{MoB: 94}}, \textcolor{red}{BoN: 72} $ | $ \textcolor{blue}{\textbf{MoB: 98}}, \textcolor{red}{BoN: 92} $ |
> |  $0.25-0.50$ | -  |  $ \textcolor{blue}{MoB: 6}, \textcolor{red}{BoN: 6} $ | $ \textcolor{blue}{MoB: 33}, \textcolor{red}{BoN: 33} $  | $ \textcolor{blue}{\textbf{MoB: 50}}, \textcolor{red}{BoN: 25} $ |
>
> In questions where the base model's success probability is at least 0.25, MoB performs better. This can be explained by the fact that number of type A questions are larger than type B questions. The positive effect of step 2, is over dominating the the negative impact of step 1 in high-accuracy reward questions as well. In contrast, in question with very small base model success probability, BoN's performance is not saturated with our current choice of $N$ (large negative impact in step 1), and also there are more type B questions than type A. This leads to BoN outperforming MoB.
>
> ## Comparison of 30 setups
>
> Now we can explain why in some setups MoB introduces more improvement than other setups. In the accuracy plots (Fig 13-17) we observe that the performance of BoN plateaus quite early in almost all benchmarks. This means the impact of step 1 should be minimal, and most of the variation among setups should come from the impact of step 2. To test this, we calculate the difference between the number of type A questions and the number of type B questions for all 30 setups. Our analysis is verified by the fact that this metric has a spearman correlation of $0.89$ with the difference between MoB accuracy and BoN accuracy.
>
> # Clarifications
>
> **(W1 and W2)** Demonstrating the capabilities of a general-purpose test-time scaling method is inherently challenging, as it involves
> choices across numerous dimensions—generative models, reward models, and benchmarks. We aimed
> for as comprehensive an evaluation as feasible by selecting generative models from diverse families and
> benchmarks spanning multiple subjects. From the RewardBench leaderboard, we intentionally selected
> strong, uncontaminated reward models, particularly those performing well on reasoning tasks. This
> combination of models and benchmarks resulted in 30 distinct experimental settings, which, in our
> view and compared to prior work, represents a broad and meaningful coverage of the space—though
> we acknowledge that alternative selections could certainly be made.
>
> Our choice of benchmarks was guided by two considerations: (1) a focus on tasks with discrete final
> answers, enabling the majority voting mechanism integral to MoB, and (2) coverage of domains where
> reasoning is critical. We also note that RewardBench2 was released after our submission deadline. However, above, we have reported math500 results for the strongest model on their leaderboard—Skywork-
> Reward-V2-Llama-3.1-8B—and these results are consistent with the overall findings we present. We
> plan to expand these updates in the paper by including results for the rest of the benchmarks and
> generative models using this reward model.
>
> **(W3)** Regarding the use of few-shot prompting versus CoT, we aimed to
> follow the common practice for each baseline setting. Our primary focus was to keep variation factors
> consistent when comparing MoB with other baselines, ensuring that differences in results reflect the
> methods themselves rather than inconsistencies in setup. The use of few-shot prompts for GSM8K
> was, in fact, unintentional—we relied on the default configurations in LM Evaluation Harness for that
> experiment. We will clarify this in the paper and ensure alignment with the implementations and
> intentions described in the original works. Lastly, we studied the general problem of selecting among a given set of outputs using scalar rewards. Li 2022 originally introduced WBoN in a more special case that we have control over the generation and more informative rewards, but our implementation of the algorithm is widely used by the community in our general problem setting as well.
>
> **(W4)**
> We have tried our best to provide a fair comparison of our method with the baselines. In our tables, intervals that contain the highest average accuracy are bolded. This is perhaps not a common strategy, and we will change it. However, it is not always in favor of MoB and is applied consistently. Lastly, the 25 out of 30 claims is based on averages.

---

> > ### Comment · Reviewer_DAFe · 2025-08-01
> > **Updated Assessment**
> >
> > I thank the authors for the detailed and thoughtful rebuttal.  They have addressed my most important concerns, and I have updated my score.  I hope the authors are able to incorporate the explanations above in their camera ready version, I found them to be clear and convincing.

---

> > > ### Author Response · Authors · 2025-08-03
> > >
> > > We appreciate the review’s response and pleased to hear that their main concerns are addressed. We will include this discussion in the camera ready version.

---

### Official Review · Reviewer_ZoLM · 2025-06-15

**Clarity:** 2
**Significance:** 3
**Originality:** 3
**Rating:** 4
**Confidence:** 4

**Summary:**

This paper proposes Majority-of-the-Bests, a method to improve Best-of-N decoding for LLMs by using bootstrapping to estimate the distribution of BoN outputs and selecting the most frequent outcome. By aggregating rewards over resampled subsets, MoB aims to mitigate errors from noisy reward models and shows consistent empirical gains across multiple benchmarks.

**Questions:**

- How do you define the Oracle final answer distribution? I am not sure how one can access the true pi_N.

**Ethical Concerns:**

["NO or VERY MINOR ethics concerns only"]

**Final Justification:**

I went through the other reviewers' questions and responses, and most of my concerns have been resolved and had a chance to have better understanding of the approach.
I now think the paper holds promise, and will adjust my scores accordingly.

**Limitations:**

There does not seem to be a separate limitations section or a societal impact section. Please add them.

**Quality:**

3

**Strengths And Weaknesses:**

# Strengths
- Conceptually simple methodology
- Consistent experimental results on a wide range of settings

# Weaknesses
- The paper frames BoN’s randomness as a problem to be mitigated. But if reward scores are noisy and correctness is inherently uncertain, is forcing the "mode" always desirable? Sometimes randomness can reflect epistemic uncertainty, not just noise.
- Bootstrapping is a statistical resampling technique that works well when the sample size is large and the statistic of interest is smooth and not too sensitive to outliers. However, this work applies bootstrapping to a relatively small sample set and attempts to approximate the extreme statistic: the maximum reward. So I am a bit suspicious about the validity and soundness of applying the bootstrapping technique.
- In Figure 2, I am curious why the authors assigned differently-sized bins for the histogram. To claim that "BoN has a significant chance of returning the correct answer but fails to do so reliably", a more fine-grained histogram is needed to demonstrate the skewness of the success probability distribution.
- The BoN itself is a computationally heavy algorithm that is not often used in practice. Although authors disclose that the CPU computation overhead is marginal, I suspect MoB will take longer to compute overall. It would be helpful to have a full comparison of computation time (latency) among methods: Best-of-1, BoN, MoB, and BoN+SC.

---

> ### Author Rebuttal · Authors · 2025-07-31
>
> We thank the reviewer for their review with multiple insightful points. Here are some thoughts and clarifications.
>
> **(W1)** This is indeed an interesting question. The uncertainty of reward and BoN output is perhaps an informative signal that can be used to modify our agent sampling budget or sampling temperature. If we are restricted to the outputs and rewards given to us, the mode might be useful. If it is correct, the agent will be correct deterministically instead of having a probability much lower. This approach has been proven widely usefully by self-consistency too. More often than not, the mode is correct and picking it almost deterministically is a better approach than sampling. Our observations confirm this for BoN output distribution.
>
> **(W2)** It is true that extreme statistics are more tricky to bootstrap. This is exactly the reason we have the more complicated approach of choosing the subsample size $m$ smaller than $N$ and carefully pick it. However, as we provide the literature, it has been studied thoroughly and has both empirical and theoretical support. The small sample size can also make the estimate difficult. The main point that makes our algorithm viable is that the accuracy of distribution estimation is not critical to the algorithm. We are only concerned with picking the mode, which can be done with rough estimations as well. Even if an error happens, we still pick a descent output as BoN already is a strong algorithm that samples from the distribution.
>
> **(W3)** The goal of that figure was to show there are significant number of question in our dataset that picking the mode is significantly different from sampling, hence motivating MoB. If the probability of the correct answer is close to 0 it will not be picked by neither of sampling or mode selection. On the other hand, if its probability is close to 1, it will be picked almost certainly by both methods. It is only in the intermediate probabilities that mode selection deterministically succeeds/fails, but sampling remains stochastic.
>
> **(W4)** The latency of all these algorithms is with orders of magnitude dominantly governed by the latency of output generation. LLM inference is far slower than the selection phase. Roughly speaking, the outputs are generated in 10s or at most 100s tokens per second and generating a typical 1000-token output will take a few seconds. For comparison, the CPU compute of MoB in our experiments was $2.6$ milliseconds, which is practically negligible. It is true that BoN has a faster selection but it is irrelevant for the whole algorithm latency.
>
> **(Q1)** In the experiments we estimate it by generating far more than $N$ outputs. The naive approach is to run Best-of-$N$ many independent times and use the empirical distribution. We use bootstrapping here to reduce this cost and use $\hat \pi_{N, 1400}$. Nonetheless, it is a very computationally heavy task and we have only done in one of the settings to provide intuitions of the algorithm. More details is provided in Appendix D.2.

---

> > ### Comment · Reviewer_ZoLM · 2025-08-02
> >
> > Dear authors,
> >
> > Thank you for your explanations.
> >
> > I also went through the other reviewers' questions and responses, and most of my concerns have been resolved and had a chance to have better understanding of your approach.
> >
> > I now think the paper holds promise, and will adjust my scores accordingly.
> >
> > Thank you.

---

> > > ### Author Response · Authors · 2025-08-03
> > >
> > > We thank the reviewer for their response and attention to our rebuttal. We are glad to hear it has clarified the ambiguities.

---

### Official Review · Reviewer_kwjQ · 2025-07-03

**Clarity:** 3
**Significance:** 3
**Originality:** 2
**Rating:** 5
**Confidence:** 3

**Summary:**

The paper proposes **Majority-of-the-Bests (MoB)**, a hyper-parameter-free alternative to Best-of-N (BoN) sampling for inference-time reasoning with LLMs.

- **Motivation.** BoN’s success probability can saturate well below 1 with imperfect rewards. If the correct answer is the most probable output of BoN, we are likely to fix the errors of BoN by doing majority voting over BoN results.

- **Key idea.** After sampling *N* candidate traces and scoring them with a reward model, MoB repeatedly bootstraps *m*-sized subsets (with replacement), keeps only the best-of-m trace in each subset, and select the mode of these “best” traces.

- **Theory.**  The authors proves that:
  - An *m-out-of-N* bootstrap consistently estimates BoN’s output distribution. That is, MoB converges to its oracle version on large N.
  - Their algorithm of adaptively select nearly optimal m with theoretical gaurantee under certain situation. Empirical results also show that adaptive m performance closely follows the optimal m variant.
- **Experiments.**  On five benchmarks, three open-weight LLMs and two reward models, MoB outperforms BoN, Self-Consistency and Weighted BoN in **25 / 30** settings; at $N = 128$ it raises accuracy by largely over BoN.
- **Practicality.**  The method adds only marginal CPU overhead; the computational-heavy sampling part remains unchanged.

**Questions:**

1. **When does MoB underperform BoN?** The appendix hints that very hard questions (base success rate <1%) may favor larger m or even plain BoN since the correct answer may no longer be the most probable one for BoN under small m. Could you add a difficulty-conditioned analysis and a practical decision rule?
2. **Larger / proprietary models.** Can you report at least a subset of results on proprietary LLMs and larger open-weight LLMs like GPT-4o, Gemini 2.5, and Qwen3-30B-A3B, perhaps on fewer datasets, to further demonstrate scalability and generalizability?
3. **Comparing with SC.** Given that MoB shares the same limitation of not supporting open-ended generation with SC, could you also add SC as an additional baseline and report MoB improvements over SC?

**Rating swing.** Demonstrating (1) a principled failure-mode analysis and (2) results on at least one >10B or proprietary model with consistent results would increase my overall score by 1 point (to 5: Accept).

**Ethical Concerns:**

["NO or VERY MINOR ethics concerns only"]

**Final Justification:**

The authors addressed my primary concern about failure modes and provided additional experimental results on proprietary models. As such, I revised my score upward.

**Limitations:**

yes

**Quality:**

3

**Strengths And Weaknesses:**

**Strengths**

- Solid theoretical analysis and a proof of bootstrap consistency; extensive evaluation across 30 settings; results reported with confidence intervals.
- Motivation is intuitive; algorithm and adaptive-*m* procedure are clearly described with helpful figures.
- MoB can serve as a drop-in replacement of BoN with marginal overhead, offering systematic gains without tuning.

**Weaknesses**

- Models stop at 9B parameters and open-weight models; no ablation on bootstrap budget *B*.
- A few typos remain; discussion of failure cases is brief and buried in the appendix.
- Share the same limited use cases as SC. Open-ended generation (summaries, codegen) is not supported, unlike BoN.
- Relies on combining previous methods (bootstrap, majority voting, BoN); originality is limited.

---

> ### Author Rebuttal · Authors · 2025-07-31
>
> We thank the reviewer for their feedback and positive review. We would like to present some clarifications and additional experimental results to address the reviewers concern.
>
> **(W1 and Q2)** Our choice of models was limited by our resources and desire to test our algorithm on many environments with reasonable confidence intervals. Nonetheless, as the reviewer suggests, some experiments on larger models will also be insightful. To address this, we here provide the results for GPT-4o-mini on MATH500 and ArmoRM reward model.
>
> |      |      Accuracy         |
> |:----:|:----------------------:|
> | BoN  |         69.4%         |
> |  SC  |         72.0%         |
> | WBoN |         72.2%         |
> | MoB  |         74.0%         |
>
> This is aligned with our observations with smaller models and show MoB outperforming the baselines. We will try to provide further results with larger models at the time of publication of the paper.
>
>
> **(W1)** Regarding ablations on $B$, we want to further clarify that the approximate distribution $\hat \pi_{m,N}$ can be calculated in close-form, and sampling from it to find its mode unnecessary. We introduced $B$ for the sake of simplicity of the main text and intuitive understanding of the algorithm. In line 181 we referred to Appendix B where the close-form is presented. Though admittedly, we have not been clear enough and other reviewers also missed this. The close-form calculation of the approximate distribution allows us to achieve the same distribution as achieved with $B = \infty$. Our implementation of the algorithm does not have a parameter $B$ at all.
>
> **(W2 and Q1)**
> We thank the reviewer for the suggestion to expand our anlysis of MoB failure modes and the conditions of its improvement, We also believe it will strengthen the paper. Towards this, we present an extensive new set of analysis that we have provided in our response to reviewer DAFe. We refer the reviewer to that response for complete details. As a summary, we further confirm our insights from Appendix C in one of realistic experiments. There are two factors that can negatively affect MoB:
>
> 1. MoB works with Best-of-$m$ instead of Best-of-$N$ for some $m < N$. This negatively affects MoB If BoN's performance is still not saturated, and we significantly benefit from a larger number of outputs ($N$ compared to $m$).
>
> 2. MoB picks the mode of output distribution of Best-of-$m$. While it is immensely helpful and is the main reason for the better overall performance of MoB compared to BoN in our benchmarks, it might be downgrade the performance if the mode is not correct. If the correct answer is not the mode, it might get a chance to be chosen when sampling, but MoB's mode selection significantly reduces this chance.
>
> In our complete discussion in our response to reviewer DAFe, we report the impact of each of these two factors for questions with various base model success probability and reward accuracy. We confirm the insight from Appendix C in the synthetic example. The first failure mode is question with very low base success probability, which the reviewer mentions here. We observe that amon these questions, either questions where the mode is incorrect are more common, or BoN can still benefit from more generations to have more correct ones. Second failure case is questions with low quality rewards. This is explained by the higher number of questions where the mode is incorrect. Overall performance of the algorithms is governed by how many questions fall into these failure modes.
>
> **(W3 and Q3)** It is true that MoB can only improve BoN when discrete output space and repetition in final answers allow majority voting. However, unlike SC which turns into random selection in absence of final answer repetition, MoB just matches BoN. We can still run MoB by setting $Z_i = f(Y_i) = Y_i$. If all $Z_i$ s are distinct, MoB's choice will be the same as Best-of-$N$, as long as $m > 1$. To see this, we can use the close-form description of $\hat\pi_{m,N}$ in line 470. Lastly, we have already included SC in all our experiments. Perhaps the reviewer could further clarify what result they believe can benefit the paper.

---

> > ### Comment · Reviewer_kwjQ · 2025-08-01
> >
> > I thank the authors for their detailed and thoughtful rebuttal. I am satisfied with the additional results and the analysis of failure modes, which addressed my primary concern. Echoing Reviewer DAFe, I strongly recommend incorporating the key explanations from the rebuttal into the camera-ready version to benefit all readers. Regarding Q2,  while the rebuttal does not fully resolve my concern about scalability & generalizability (since there are rumors that GPT-4o-mini is also ~8B), I still appreciate the additional results provided. For Q3, I'm suggesting a more rigorous evaluation baseline. Given that MoB only applies to settings where both BoN and SC are viable, its improvement should also be reported against the stronger of the two baselines for each specific setting. I have raised my score as promised to reflect the improvements.

---

> > > ### Author Response · Authors · 2025-08-03
> > >
> > > We are glad that the rebuttal has addressed reviewer’s main concerns. We will indeed incorporate the new analysis given in the rebuttal and the reviewer‘s suggestions in the camera ready version.

---

### Official Review · Reviewer_ufGg · 2025-07-06

**Clarity:** 3
**Significance:** 3
**Originality:** 2
**Rating:** 4
**Confidence:** 4

**Summary:**

The paper proposes "Majority of the Bests (MoB)" a simple method to improve best-of-n sampling with LLMs using  bootstrapping. The proposed method works by sampling multiple times from generated LLM outputs, selecting the highest rewarded output, followed by selecting the majority answer from the filtered outputs. The paper provides both theoretical and experimental support for superiority of their method. When evaluated on multiple datasets, 3 generator models and 2 different reward models, the method outperforms both BoN and weighted BoN.

**Questions:**

Please See the Weaknesses Above.

Additionally:
1. The results in Table 1, Row: WBoN and Column: CSQA are abnormally low. Is there a typo?
2. I particularly like the Figure 6 showcasing how Accuracy varies with number of generations. Form graph it seems, the performance difference is particularly higher for smaller N. Could authors provide results for other setups, and plot performance difference as a function of N? The analysis might further strengthen the paper.

**Ethical Concerns:**

["NO or VERY MINOR ethics concerns only"]

**Final Justification:**

The authors have addressed my concerns thoroughly, and overall, the contribution of the paper is valuable. However, at the same time, the results are not consistently better than WoB (eg, with SkyWork RM), highlighting that the proposed method may not be a universal replacement.

**Limitations:**

yes

**Quality:**

3

**Strengths And Weaknesses:**

## Strengths:
1. The paper proposes a simple method to improve the performance of LLMs, while showcasing strong results across a broad range of datasets and models.
2. The method can act as a replacement for the widely used Weighted Best-of-N method across the domains tested.
3. The paper is well-presented and easy to follow.

---

## Weaknesses
1. The MoB method works based on the assumption that reward model outputs are noisy. While generally true, the paper does not compare against the latest strong reward models. If the authors could show results with them (e.g., by choosing the strongest reward model from RewardBench on respective domains), the effectiveness of MoB could be further strengthened.
2. In the text and tables, the authors continuously emphasize comparison with BoN instead of WBoN (e.g., the improvement row in Table 2). Further, the authors claim "MoB is a drop-in replacement for BoN". However, this framing is arguably unfair, since evaluations are a.) only done on tasks with a discrete output space and b.) majority is calculated over final answers and not full outputs.
3.  There is limited analysis in the paper. For instance, the authors could add what are some failure modes of the method, on which questions the methods fail, how scaling RM size changes the performance, or the effects of hyperparameters. The complete results for the stated "30 setups" are also not presented.
4. The paper claims the method requires no hyperparameter tuning; however, both variables q and B have to be chosen, and an analysis studying their effects is missing.
5. Scaling the number of model outputs is one dimension, while scaling the number of outputs from generative reward models (which are often used) is another. Comparing the effect, and how MoB can be used with generative reward models, would be an interesting study.
6. The paper claims the method is computationally efficient. However, it may still require significant CPU resources because of the large value of B (10,000), which might increase latency and cost in real-world usage. Results showing actual wall time and CPU usage are missing.
7.  The paper lacks a dedicated Related Work section. While a minor issue, omitting one makes it harder to understand the work's novelty.
---

Overall, the proposed method is promising with some exciting results. However, a more thorough analysis is needed to further validate the claims.

---

> ### Author Rebuttal · Authors · 2025-07-31
>
> We appreciate the reviewer's close attention to our work and thank them for their time and thorough review. We present some clarifications and additional experimental results to hopefully address the reviewers concerns.
>
> **(W1)** We appreciate the reviewer's suggestion. Here are the results on MATH500 benchmark using Skywork-Reward-V2-Llama-3.1-8B reward model. This model is currently ranked first on RewardBench overal ranking and second in the math ranking.
>
> |      |  Llama3.1-8b-instruct  |  Gemma2-9b-it  |  Qwen2.5-3b-instruct  |
> |:----:|:----------------------:|:--------------:|:---------------------:|
> | BoN  |         54.50%         |     55.85%     |        63.35%         |
> |  SC  |         60.65%         |     52.90%     |        66.40%         |
> | WBoN |         65.05%         |     57.95%     |        68.80%         |
> | MoB  |         63.95%         |     59.45%     |        69.70%         |
>
> We see that MoB's again provides the best or the second best performance. It is important to note that the dependence on reward imperfection is not a sign that MoB will become obsolete with better reward models, or it lacks any merit. First, as models progress, we will aim to solve more and more difficult tasks that will keep challenging the models. The newer and stronger reward models will again prvide imperfect models in the newer more difficult tasks. Second, there is a significant cost in using larger models. MoB provides an unarguable benefit by allowing the use of cheaper models that provide lower quality rewards to save costs. The chosen reward models were the SOTA in reasoning performance on RewardBench among their size classes at the time of submission.
>
> **(W2)** We understand the reviewer's point of view and perhaps the paper needs some clarification. The reason we compare MoB to BoN is because of its algorithmic similarity to it. We view MoB as an algorithm that picks the mode of BoN output distribution instead of sampling from it. Also it is important to note that if the output space is not discrete (all $Z_i$ s are distinct), BoN, WBoN, and MoB all become equivalent (See equation at line 470 in appendix for justification for MoB-BoN equivalence). It is true that MoB's improvement is only in the cases of discrete outputs. and they are equivalent otherwise.
> As a brief comparison with WBoN, MoB has a higher accuracy in 27 out of 30 setups (ignoring confidence intervals).
>
> **(W3)** Indeed, we agree that a breakdown of for questions the MoB is the most helpful is a valuable insight. In our respponse to DAFe, we have included an extensive study of this subject. We categorize questions based on the quality of output generation and rewards and describe the behavior of two factors that together determine how MoB performs compared to BoN. Lastly, the results for all 30 setups are already included in our supplementary material due to the space constraints of the main text.
>
> **(W4)** We still want to emphasize that neither of $q$ and $B$ need to be tuned. Regarding $B$, in fact there is no choice of $B$ in our implementation! As we briefly mention in line 180 (which perhaps needs to be more clear) and explain in detail in Appendix B, $\hat \pi_{m,N}$ has a close-form solution and there is no need to estimate it through $B$ samples. Using this close form, we operate equivalent to $B = \infty$ as also mentioned in implementation details in line 522 of supplementary material. Even if $B$ is used for Monte Carlo estimation, it is rarely discussed in any bootstrap papers as it just needs to be sufficiently large. Also for $q$, there is literature on the insignificance of its choice (c.f. line 283). To further test this in the following table, we provide the ablation results on the value of $q$ on Llama3.1-8b-instruct and ArmoRM across benchmarks, which confirms the negligible importance of the choice of $q$.
>
>
> |      $q$      |  $0.40$  |  $0.50$  |  $0.60$  |  $0.70$  |  $0.80$  |  $0.90$  |
> |:-------------:|:--------:|:--------:|:--------:|:--------:|:--------:|:--------:|
> |    MATH500    |  61.85%  |  63.00%  |  63.15%  |  62.20%  |  62.45%  |  60.70%  |
> | MMLU-Pro-Math |  66.60%  |  66.85%  |  66.75%  |  66.60%  |  67.10%  |  66.35%  |
> | MMLU-Pro-Chem |  56.75%  |  57.40%  |  57.85%  |  57.15%  |  56.95%  |  56.35%  |
> |     GSM8k     |  91.60%  |  91.55%  |  91.55%  |  91.85%  |  91.80%  |  91.85%  |
> |     CSQA      |  77.45%  |  77.40%  |  77.30%  |  77.25%  |  77.35%  |  77.30%  |
>
>
> **(W5)**
>  We agree with the reviewer that further studies on generative reward models will be insightful. We hope the results with Skywork reward model addresses the reviewers concern.
>
> **(W6)**
> We again refer the reviewer to Appendix B where we introduce a close-form calculation of $\hat \pi_{m,N}$ that allows us to operate equivalent to $B=\infty$ in $O(N\log N)$ compute complexity. Our implementation of MoB took an average of 2.6 milliseconds on a laptop CPU. This is irrelevant compared the latency in generating the outputs.
>
> **(Q1)** This is the result of one of the weaknesses of WBoN that limits its applicability to positive rewards. It seems like GRM reward model is outputing negative rewards in the CSQA benchmarks. We are not aware of a standard way to fix this. For example naively adding a constant to the rewards, turns WBoN to SC.
>
> **(Q2)** Absolutely. The plots for all setups are provided in the supplementary material (Figures 13-17).

---

> > ### Comment · Reviewer_ufGg · 2025-08-06
> >
> > I thank the authors for their detailed rebuttal. The discussion and added results are all convincing. I have increased my score accordingly.
> > I hope the authors include the above discussion, especially for W1 and W4, in the revision. Also, in the current version, supplementary material was often not referenced at appropriate places in the main paper, which could be fixed in the revised version.

---

> > > ### Author Response · Authors · 2025-08-06
> > >
> > > We thank the reviewer for their attention to our rebuttal and happy to hear their main concerns were addressed. As the reviewer suggests, we will ensure the new discussions and clarifications are applied to the future revisions of the paper.

---

### Decision · Program_Chairs · 2025-09-17

**Decision:**

Accept (poster)

**Comment:**

This paper introduces a simple method to improve the Best-of-N (BoN) inference strategy by leveraging bootstrapping to find the most frequent answer. The approach is well-motivated by the problem of noisy reward models and is supported by both theoretical consistency guarantees and experiments across numerous benchmarks and models, where it consistently outperforms established baselines. The reviewers engaged in a thorough discussion, and the authors provided a detailed rebuttal, including an analysis of the method's performance under different conditions and additional experiments with stronger reward models and proprietary LLMs. The consensus is that the paper presents a valuable and well-validated contribution to improving inference-time computation for LLMs.